# Taming Diffusion Times in Score-based Generative Models: Trade-offs and Solutions

## Abstract

Score-based diffusion models are a class of generative models whose dynamics is described by stochastic differential equations that map noise into data. While recent works have started to lay down a theoretical foundation for these models, an analytical understanding of the role of the diffusion time $T$ is still lacking. Current best practice advocates for a large $T$ to ensure that the forward dynamics brings the diffusion sufficiently close to a known and simple noise distribution; however, a smaller value of $T$ should be preferred for a better approximation of the score-matching objective and higher computational efficiency. Starting from a variational interpretation of diffusion models, in this work we quantify this trade-off, and suggest a new method to improve quality and efficiency of both training and sampling, by adopting smaller diffusion times. Indeed, we show how an auxiliary model can be used to bridge the gap between the ideal and the simulated forward dynamics, followed by a standard reverse diffusion process. Empirical results support our analysis; for image data, our method is competitive w.r.t. the state-of-the-art, according to standard sample quality metrics and log-likelihood.

## 1 Introduction

Diffusion-based generative models (Sohl-Dickstein et al., 2015; Song & Ermon, 2019; Song et al., 2021c; Vahdat et al., 2021; Kingma et al., 2021; Ho et al., 2020; Song et al., 2021a) have recently gained popularity due to their ability to synthesize high-quality audio (Kong et al., 2021; Lee et al., 2022b), image Dhariwal & Nichol (2021); Nichol & Dhariwal (2021) and other data modalities Tashiro et al. (2021), outperforming known methods based on Generative Adversarial Networks (GANs) (Goodfellow et al., 2014), normalizing flows (NFs) (Kingma et al., 2016) or Variational Autoencoders (VAEs) and Bayesian autoencoders (BAEs) (Kingma & Welling, 2014; Tran et al., 2021).

Diffusion models learn to generate samples from an unknown density $p_{data}$ by reversing a *diffusion process* which transforms the distribution of interest into noise. The forward dynamics injects noise into the data following a diffusion process that can be described by a Stochastic Differential Equation (SDE) of the form,

$$\mathrm{d}\boldsymbol{x}_t = \boldsymbol{f}(\boldsymbol{x}_t, t)\mathrm{d}t + g(t)\mathrm{d}\boldsymbol{w}_t \quad \text{with} \quad \boldsymbol{x}_0 \sim p_{data} , \tag{1}$$

where $\boldsymbol{x}_t$ is a random variable at time $t$, $\boldsymbol{f}(\cdot, t)$ is the *drift term*, $g(\cdot)$ is the *diffusion term* and $\boldsymbol{w}_t$ is a *Wiener process* (or Brownian motion). We will also consider a special class of linear SDEs, for which the drift term is decomposed as $f(\boldsymbol{x}_t, t) = \alpha(t)\boldsymbol{x}_t$ and the diffusion term is independent of $\boldsymbol{x}_t$. This class of parameterizations of SDEs is known as *affine* and it admits analytic solutions. We denote the time-varying probability density by $p(\boldsymbol{x}, t)$, where by definition $p(\boldsymbol{x}, 0) = p_{data}(\boldsymbol{x})$, and the conditional on the initial condition $\boldsymbol{x}_0$ by $p(\boldsymbol{x}, t \,|\, \boldsymbol{x}_0)$. The forward SDE is usually considered for a sufficiently long *diffusion time $T$*, leading to the density $p(\boldsymbol{x}, T)$. In principle, when $T \to \infty$, $p(\boldsymbol{x}, T)$ converges to Gaussian noise, regardless of initial conditions.

For generative modeling purposes, we are interested in the inverse dynamics of such process, i.e., transforming samples of the noisy distribution $p(\boldsymbol{x}, T)$ into $p_{data}(\boldsymbol{x})$. Formally, such dynamics can be obtained by considering the solutions of the inverse diffusion process (Anderson, 1982),

$$\mathrm{d}\boldsymbol{x}_t = \left[ -\boldsymbol{f}(\boldsymbol{x}_t, t') + g^2(t')\boldsymbol{\nabla} \log p(\boldsymbol{x}_t, t') \right] \mathrm{d}t + g(t')\mathrm{d}\boldsymbol{w}_t , \tag{2}$$

where $t' \stackrel{\text{def}}{=} T - t$, with the inverse dynamics involving a new Wiener process. Given $p(\boldsymbol{x}, T)$ as the initial condition, the solution of Eq. (2) after a *reverse diffusion time* $T$, will be distributed as $p_{data}(\boldsymbol{x})$. We refer to the density associated to the backward process as $q(\boldsymbol{x}, t')\overline{q(\boldsymbol{x}, t)}$. The simulation of the backward process is referred to as *sampling* and, differently from the forward process, this process is not *affine* and a closed form solution is out of reach. PM: in Eq. (2), should we at least say that the Weiner process is in reverse time?

**Practical considerations on diffusion times.** In practice, diffusion models are challenging to work with (Song et al., 2021c). Indeed, a direct access to the true *score* function $\boldsymbol{\nabla} \log p(\boldsymbol{x}_t, t)$ required in the dynamics of the reverse diffusion is unavailable. This can be solved by approximating it with a parametric function $\boldsymbol{s}_{\boldsymbol{\theta}}(\boldsymbol{x}_t, t)$, e.g., a neural network, which is trained using the following loss function,

$$\mathcal{L}(\boldsymbol{\theta}) = \int_0^T \mathbb{E}_{\sim(1)} \lambda(t) \|\boldsymbol{s}_{\boldsymbol{\theta}}(\boldsymbol{x}_t, t) - \boldsymbol{\nabla} \log p(\boldsymbol{x}_t, t \,|\, \boldsymbol{x}_0)\|^2, \tag{3}$$

$$\cancel{\mathcal{L}(\boldsymbol{\theta}) = T\, \mathbb{E}_{p(t)} \mathbb{E}_{\sim(1)} \lambda(t) \|\boldsymbol{s}_{\boldsymbol{\theta}}(\boldsymbol{x}_t, t) - \boldsymbol{\nabla} \log p(\boldsymbol{x}_t, t \,|\, \boldsymbol{x}_0)\|^2,}$$

where $\lambda(t)$ is a positive weighting factor and the notation $\mathbb{E}_{\sim(1)}$ means that the expectation is taken with respect to the random process $\boldsymbol{x}_t$ in Eq. (1): for a generic function $h$, $\mathbb{E}_{\sim(1)}[h(\boldsymbol{x}_t, \boldsymbol{x}_0, t)] = \int h(\boldsymbol{x}, \boldsymbol{z}, t) p(\boldsymbol{x}, t \,|\, \boldsymbol{z}) p_{data}(\boldsymbol{z}) \mathrm{d}\boldsymbol{x} \mathrm{d}\boldsymbol{z}$. $\cancel{p(t) = \mathcal{U}(0, T)}$. This loss, usually referred to as *score matching loss*, is the cost function considered in Song et al. (2021b) (Eq. (4) ). The condition $\lambda(t) = g(t)^2$, adopted in this work, is referred to a *likelihood reweighting*. Due to the affine property of the drift, the term $p(\boldsymbol{x}_t, t \,|\, \boldsymbol{x}_0)$ is analytically known and normally distributed for all $t$ (expression available in Table 1, and in Särkkä & Solin (2019)). ~~Note also that we will refer to $\lambda$ as the *likelihood reweighting* factor when $\lambda(t) = g(t)^2$~~ (Song et al., 2021b). Intuitively, the estimation of the *score* is akin to a denoising objective, which operates in a challenging regime. Later we will quantify precisely the difficulty of learning the *score*, as a function of increasing diffusion times.

While the forward and reverse diffusion processes are valid for all $T$, the noise distribution $p(\boldsymbol{x}, T)$ is analytically known only when the diffusion time is $T \to \infty$. To overcome this problem, the common solution is to replace $p(\boldsymbol{x}, T)$ with a simple distribution $p_{noise}(\boldsymbol{x})$ which, for the classes of SDEs we consider in this work, is a Gaussian distribution.

In the literature, the discrepancy between $p(\boldsymbol{x}, T)$ and $p_{noise}(\boldsymbol{x})$ has been neglected, under the informal assumption of a sufficiently large diffusion time. Unfortunately, while this approximation seems a valid approach to simulate and generate samples, the reverse diffusion process starts from a different initial condition $q(\boldsymbol{x}, 0)$ and, as a consequence, it will converge to a solution $q(\boldsymbol{x}, T)$ that is different from the true $p_{data}(\boldsymbol{x})$. Later, we will expand on the error introduced by this approximation, but for illustration purposes Fig. 1 shows quantitatively this behavior for a simple 1D toy example $p_{data}(\boldsymbol{x}) = \pi \mathcal{N}(1, 0.1^2) + (1 - \pi) \mathcal{N}(3, 0.5^2)$, with $\pi = 0.3$: when $T$ is small, the distribution $p_{noise}(\boldsymbol{x})$ is very different from $p(\boldsymbol{x}, T)$ and samples from $q(\boldsymbol{x}, T)$ exhibit very low likelihood of being generated from $p_{data}(\boldsymbol{x})$.

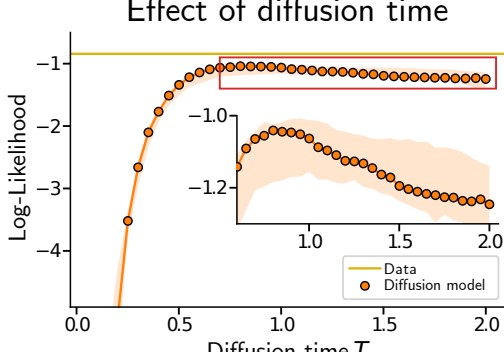

**Figure 1:** Effect of $T$ on a toy model: low diffusion times are detrimental for sample quality (likelihood of 1024 samples as median and 95 quantile, on 8 random seeds).

Crucially, Fig. 1 (zoomed region) illustrates an unknown behavior of diffusion models, which we unveil in our analysis. The right balance between efficient *score* estimation and sampling quality can be achieved by diffusion times that are smaller than common best practices. This is a key observation we explore in our work.

**Contributions.** An appropriate choice of the diffusion time $T$ is a key factor that impacts training convergence, sampling time and quality. On the one hand, the approximation error introduced by considering initial conditions for the reverse diffusion process drawn from a simple distribution $p_{noise}(\boldsymbol{x}) \neq p(\boldsymbol{x}, T)$ increases when $T$ is small. This is why the current best practice is to choose a sufficiently long diffusion time. On the other hand, training convergence of the *score* model $\boldsymbol{s_\theta}(\boldsymbol{x}_t, t)$ becomes more challenging to achieve with a large $T$, which also imposes extremely high computational costs **both** for training and for sampling. This would suggest to choose a smaller diffusion time. Given the importance of this problem, in this work we set off to study—for the first time—the existence of suitable operating regimes to strike the right balance between computational efficiency and model quality. The main contributions of this work are the following.

**Contribution 1:** In § 2 ~~we provide a new characterization of score-based diffusion models, which allows us to obtain a formal understanding of the impact of the diffusion time $T$. We do so by introducing a novel decomposition of the evidence lower bound (ELBO), which emphasizes the roles of (i) the discrepancy between the "ending" distribution of the diffusion and the "starting" distribution of the reverse diffusion processes, and (ii) of the *score* matching objective. This allows us to claim the existence of an optimal diffusion time, and it provides, for the first time, a formal assessment of the current best practice for selecting $T$.~~ we use an ELBO decomposition which allows us to study the impact of the diffusion time $T$. This ELBO decomposition emphasizes the roles of (i) the discrepancy between the "ending" distribution of the diffusion and the "starting" distribution of the reverse diffusion processes, and (ii) of the *score* matching objective. Crucially, our analysis does not rely on assumptions on the quality of the score models. We explicitly study the existence of a trade-off and explore experimentally, for the first time, current approaches for selecting the diffusion time $T$.

**Contribution 2:** In § 3 we propose a novel method to improve *both* training and sampling efficiency of diffusion-based models, while maintaining high sample quality. Our method introduces an auxiliary distribution, allowing us to transform the simple "starting" distribution of the reverse process used in the literature so as to minimize the discrepancy to the "ending" distribution of the forward process. Then, a standard reverse diffusion can be used to closely match the data distribution. Intuitively, our method allows to build "bridges" across multiple distributions, and to set $T$ toward the advantageous regime of small diffusion times.

In addition to our methodological contributions, in § 4, we provide experimental evidence of the benefits of our method, in terms of sample quality and log likelihood. Finally, we conclude in § 5.

**Related Work.** A concurrent work by Zheng et al. (2022) presents an empirical study of a truncated diffusion process, but lacks a rigorous analysis, and a clear justification for the proposed approach. Recent attempts by Lee et al. (2022b) to optimize $p_{noise}$, or the proposal to do so (Austin et al., 2021) have been studied in different contexts. Related work focus primarily on improving sampling efficiency, using a wide array of techniques. Sample generation times can be drastically reduced considering adaptive step-size integrators (Jolicoeur-Martineau et al., 2021). Other popular choices are based on merging multiple steps of a pretrained model through distillation techniques (Salimans & Ho, 2022) or by taking larger sampling steps with GANs (Xiao et al., 2022). Approaches closer to ours *modify* the SDE, or the discrete time processes, to obtain inference efficiency gains. In particular, Song et al. (2021a) considers implicit non-Markovian diffusion processes, while Watson et al. (2021) changes the diffusion processes by optimal scheduling selection and Dockhorn et al. (2022) considers overdamped SDEs. Finally, hybrid techniques combining VAEs and diffusion models (Vahdat et al., 2021) or simple auto encoders and diffusion models (Rombach et al., 2022), have positive effects on training and sampling times.

## 2 ~~A new ELBO decomposition and a tradeoff on diffusion time~~ Exploring a tradeoff on diffusion time

The dynamics of a diffusion model can be studied through the lens of variational inference, which allows us to bound the (log-)likelihood using an evidence lower bound (ELBO) (Huang et al., 2021).

~~Our interpretation~~ The interpretation we consider in this work (see also Song et al. (2021b), Thm. 1) emphasizes the two main factors affecting the quality of sample generation: an imperfect *score*, and a mismatch, measured in terms of the Kullback-Leibler (KL) divergence, between the noise distribution $p(\boldsymbol{x}, T)$ of the forward process and the distribution $p_{noise}$ used to initialize the backward process.

## 2.1 Preliminaries: the ELBO decomposition

Our goal is to study the quality of the generated data distribution as a function of the diffusion time $T$. Then, instead of focusing on the log-likelihood bounds for single datapoints $\log q(\boldsymbol{x}, T)$, we consider the average over the data distribution, i.e. the *cross-entropy* $\mathbb{E}_{p_{data}(\boldsymbol{x})} \log q(\boldsymbol{x}, T)$. By rewriting the $\mathcal{L}_{\mathrm{ELBO}}$ derived in Huang et al. (2021, Eq. (25)) (details of the steps in the Appendix), we have that

$$\mathbb{E}_{p_{data}(\boldsymbol{x})} \log q(\boldsymbol{x}, T) \geq \mathcal{L}_{\mathrm{ELBO}}(\boldsymbol{s_\theta}, T) = \mathbb{E}_{\sim(1)} \log p_{noise}(\boldsymbol{x}_T) - I(\boldsymbol{s_\theta}, T) + R(T). \tag{4}$$

where $R(T) = \frac{1}{2} \int\limits_{t=0}^{T} \mathbb{E}_{\sim(1)} \left[ g^2(t) \|\boldsymbol{\nabla} \log p(\boldsymbol{x}_t, t \mid \boldsymbol{x}_0)\|^2 - 2\boldsymbol{f}^\top(\boldsymbol{x}_t, t) \boldsymbol{\nabla} \log p(\boldsymbol{x}_t, t \mid \boldsymbol{x}_0) \right] \mathrm{d}t$, and $I(\boldsymbol{s_\theta}, T) = \frac{1}{2} \int\limits_{t=0}^{T} g^2(t) \mathbb{E}_{\sim(1)} \left[ \|\boldsymbol{s_\theta}(\boldsymbol{x}_t, t) - \boldsymbol{\nabla} \log p(\boldsymbol{x}_t, t \mid \boldsymbol{x}_0)\|^2 \right] \mathrm{d}t$ is equal to the loss term Eq. (3) when $\lambda(t) = g^2(t)$.

Note that $R(T)$ depends neither on $\boldsymbol{s_\theta}$ nor on $p_{noise}$, while $I(\boldsymbol{s_\theta}, T)$, or an equivalent reparameterization (Huang et al., 2021; Song et al., 2021b, Eq. (1)), is used to learn the approximated *score*, by optimization of the parameters $\boldsymbol{\theta}$. It is then possible to show that

$$I(\boldsymbol{s_\theta}, T) \geq \underbrace{I(\boldsymbol{\nabla} \log p, T)}_{\stackrel{\mathrm{def}}{=} K(T)} = \frac{1}{2} \int\limits_{t=0}^{T} g^2(t) \mathbb{E}_{\sim(1)} \left[ \|\boldsymbol{\nabla} \log p(\boldsymbol{x}_t, t) - \boldsymbol{\nabla} \log p(\boldsymbol{x}_t, t \mid \boldsymbol{x}_0)\| \right]^2 \mathrm{d}t. \tag{5}$$

Note that the term $K(T) = I(\boldsymbol{\nabla} \log p, T)$ does not depend on $\boldsymbol{\theta}$. Consequently, we can define $\mathcal{G}(\boldsymbol{s_\theta}, T) = I(\boldsymbol{s_\theta}, T) - K(T)$ (see Appendix for details), where $\mathcal{G}(\boldsymbol{s_\theta}, T)$ is a positive term that we call the *gap* term, accounting for the practical case of an imperfect *score*, i.e. $\boldsymbol{s_\theta}(\boldsymbol{x}_t, t) \neq \boldsymbol{\nabla} \log p(\boldsymbol{x}_t, t)$. It also holds that

$$\mathbb{E}_{\sim(1)} \log p_{noise}(\boldsymbol{x}_T) = \int \left[ \log p_{noise}(\boldsymbol{x}) - \log p(\boldsymbol{x}, T) + \log p(\boldsymbol{x}, T) \right] p(\boldsymbol{x}, T) \mathrm{d}\boldsymbol{x} =$$
$$= \mathbb{E}_{\sim(1)} \log p(\boldsymbol{x}_T, T) - \mathrm{KL} \left[ \log p(\boldsymbol{x}, T) \parallel p_{noise}(\boldsymbol{x}) \right]. \tag{6}$$

Therefore, we can substitute the cross-entropy term $\mathbb{E}_{\sim(1)} \log p_{noise}(\boldsymbol{x}_T)$ of ~~rewrite~~ the ELBO in Eq. (4) ~~as to~~ obtain

$$\mathbb{E}_{p_{data}(\boldsymbol{x})} \log q(\boldsymbol{x}, T) \geq -\mathrm{KL} \left[ p(\boldsymbol{x}, T) \parallel p_{noise}(\boldsymbol{x}) \right] + \mathbb{E}_{\sim(1)} \log p(\boldsymbol{x}_T, T) - K(T) + R(T) - \mathcal{G}(\boldsymbol{s_\theta}, T). \tag{7}$$

Before concluding our derivation it is necessary to introduce an important Proposition ~~observation~~ (formal proof in Appendix), where we show how to combine different terms of Eq. (7) into the negative entropy term $\mathbb{E}_{p_{data}(\boldsymbol{x})} \log p_{data}(\boldsymbol{x})$.

**Proposition 1.** *Given the stochastic dynamics defined in Eq. (1), it holds that*

$$\mathbb{E}_{\sim(1)} \log p(\boldsymbol{x}_T, T) - K(T) + R(T) = \mathbb{E}_{p_{data}(\boldsymbol{x})} \log p_{data}(\boldsymbol{x}). \tag{8}$$

Finally, we can now bound the value of $\mathbb{E}_{p_{data}(\boldsymbol{x})} \log q(\boldsymbol{x}, T)$ as

$$\mathbb{E}_{p_{data}(\boldsymbol{x})} \log q(\boldsymbol{x}, T) \geq \underbrace{\mathbb{E}_{p_{data}(\boldsymbol{x})} \log p_{data}(\boldsymbol{x}) - \mathcal{G}(\boldsymbol{s_\theta}, T) - \mathrm{KL} \left[ p(\boldsymbol{x}, T) \parallel p_{noise}(\boldsymbol{x}) \right]}_{\mathcal{L}_{\mathrm{ELBO}}(\boldsymbol{s_\theta}, T)}. \tag{9}$$

Eq. (9) clearly emphasizes the roles of an approximate score function, through the gap term $\mathcal{G}(\cdot)$, and the discrepancy between the noise distribution of the forward process, and the initial distribution of the reverse

**Table 1:** Two main families of diffusion processes, where $\sigma^2(t) = \left(\frac{\sigma_{max}^2}{\sigma_{min}^2}\right)^t$ and $\beta(t) = \beta_0 + (\beta_1 - \beta_0)t$

| | Diffusion process | $p(\boldsymbol{x}_t, t \mid \boldsymbol{x}_0) = \mathcal{N}(\boldsymbol{m}, s\boldsymbol{I})$ | $p_{noise}(\boldsymbol{x})$ |
|---|---|---|---|
| Variance Exploding | $\alpha(t) = 0$, $g(t) = \sqrt{\frac{d\sigma^2(t)}{dt}}$ | $\boldsymbol{m} = \boldsymbol{x}_0$, $s = \sigma^2(t) - \sigma^2(0)$ | $\mathcal{N}(\boldsymbol{0}, (\sigma^2(T) - \sigma^2(0))\boldsymbol{I})$ |
| Variance Preserving | $\alpha(t) = -\frac{1}{2}\beta(t)$, $g(t) = \sqrt{\beta(t)}$ | $\boldsymbol{m} = e^{-\frac{1}{2}\int_0^t \beta(d\tau)} \boldsymbol{x}_0$, $s = 1 - e^{-\int_0^t \beta(d\tau)}$ | $\mathcal{N}(\boldsymbol{0}, \boldsymbol{I})$ |

process, through the KL term. The (negative) entropy term $\mathbb{E}_{p_{data}(\boldsymbol{x})} \log p_{data}(\boldsymbol{x})$, which is constant w.r.t $T$ and $\boldsymbol{\theta}$, is the best value achievable by the ELBO. Indeed, by rearranging Eq. (9), $\text{KL}\left[q(\boldsymbol{x}, T) \parallel p_{data}(\boldsymbol{x})\right] \leq \mathcal{G}(\boldsymbol{s_\theta}, T) + \text{KL}\left[p(\boldsymbol{x}, T) \parallel p_{noise}(\boldsymbol{x})\right]$. Optimality is achieved when i) we have perfect *score* matching and ii) the initial conditions for the reverse process are ideal, i.e. $q(\boldsymbol{x}, 0) = p(\boldsymbol{x}, T)$ ~~In the ideal case of perfect score matching, the ELBO in Eq. (9) is attained with equality. If, in addition, the initial conditions for the reverse process are ideal, i.e. $q(\boldsymbol{x}, 0) = p(\boldsymbol{x}, T)$, then the results in Anderson (1982) allow us to claim that $q(\boldsymbol{x}, T) = p_{data}(\boldsymbol{x})$.~~

Next, we show the existence of a tradeoff: the KL decreases with $T$, while the gap increases with $T$.

## 2.2 The tradeoff on diffusion time

We begin by showing that the KL term in Eq. (9) decreases with the diffusion time $T$, which induces to select large $T$ to maximize the ELBO. We consider the two main classes of SDEs for the forward diffusion process defined in Eq. (1): SDEs whose steady state distribution is the standard multivariate Gaussian, referred to as *Variance Preserving* (VP), and SDEs without a stationary distribution, referred to as *Variance Exploding* (VE), which we summarize in Table 1. The standard approach to generate new samples relies on the backward process defined in Eq. (2), and consists in setting $p_{noise}$ in agreement with the form of the forward process SDE. The following result bounds the discrepancy between the noise distribution $p(\boldsymbol{x}, T)$ and $p_{noise}$.

**Lemma 1.** *For the classes of SDEs considered (Table 1), the discrepancy between $p(\boldsymbol{x}, T)$ and the $p_{noise}(\boldsymbol{x})$ can be bounded as follows.*

*For Variance Preserving SDEs, it holds that:* $\text{KL}\left[p(\boldsymbol{x}, T) \parallel p_{noise}(\boldsymbol{x})\right] \leq C_1 \exp\left(-\int_0^T \beta(t)dt\right)$.

*For Variance Exploding SDEs, it holds that:* $\text{KL}\left[p(\boldsymbol{x}, T) \parallel p_{noise}(\boldsymbol{x})\right] \leq C_2 \frac{1}{\sigma^2(T) - \sigma^2(0)}$.

Our proof uses results from Villani (2009), the logarithmic Sobolev Inequality and Gronwall inequality (see Appendix for details). The consequence of Lemma 1 is that to maximize the ELBO, the diffusion time $T$ should be as large as possible (ideally, $T \to \infty$), such that the KL term vanishes. This result is in line with current practices for training score-based diffusion processes, that argue for sufficiently long diffusion times (De Bortoli et al., 2021). Our analysis, on the other hand, highlights how this term is only one of the two contributions to the ELBO.

Now, we focus our attention on studying the behavior of the second component, $\mathcal{G}(\cdot)$. Before that, we define a few quantities that allow us to write the next important result.

**Definition 1.** *We define the optimal score $\widehat{\boldsymbol{s}_\theta}$ for any diffusion time $T$, as the score obtained using parameters that minimize $I(\boldsymbol{s_\theta}, T)$. Similarly, we define the optimal score gap $\mathcal{G}(\widehat{\boldsymbol{s}_\theta}, T)$ for any diffusion time $T$, as the gap attained when using the optimal score.*

**Lemma 2.** *The optimal score gap term $\mathcal{G}(\widehat{\boldsymbol{s}_\theta}, T)$ is a non-decreasing function in $T$. That is, given $T_2 > T_1$, and $\boldsymbol{\theta}_1 = \arg\min_{\boldsymbol{\theta}} I(\boldsymbol{s_\theta}, T_1), \boldsymbol{\theta}_2 = \arg\min_{\boldsymbol{\theta}} I(\boldsymbol{s_\theta}, T_2)$, then $\mathcal{G}(\boldsymbol{s}_{\boldsymbol{\theta}_2}, T_2) \geq \mathcal{G}(\boldsymbol{s}_{\boldsymbol{\theta}_1}, T_1)$.*

The proof (see Appendix) is a direct consequence of the definition of $\mathcal{G}$ and the optimality of the score. Note that Lemma 2 does not imply that $\mathcal{G}(\boldsymbol{s}_{\boldsymbol{\theta}_a}, T_2) \geq \mathcal{G}(\boldsymbol{s}_{\boldsymbol{\theta}_b}, T_1)$ holds for generic parameters $\boldsymbol{\theta}_a, \boldsymbol{\theta}_b$.

### 2.3 Is there an optimal diffusion time?

While diffusion processes are generally studied for $T \to \infty$, for practical reasons, diffusion times in score-based models have been arbitrarily set to be "sufficiently large" in the literature. Here we formally argue, for the first time, about the existence of an optimal diffusion time, which strikes the right balance between the gap $\mathcal{G}(\cdot)$ and the KL terms of the ELBO in Eq. (9).

Before proceeding any further, we clarify that our final objective is not to find and use an optimal diffusion time. Instead, our result on the existence of optimal diffusion times (which can be smaller than the ones set by than popular heuristics) serves the purpose of motivating the choice of small diffusion times, which however call for a method to overcome approximation errors.

**Proposition 2.** *Consider the ELBO decomposition in Eq. (9). We study it as a function of time $T$, and seek its optimal argument $T^\star = \arg\max_T \mathcal{L}_{\mathrm{ELBO}}(\widehat{\boldsymbol{s}}_{\boldsymbol{\theta}}, T)$. Then, the optimal diffusion time $T^\star \in \mathbb{R}^+$, and thus not necessarily $T^\star = \infty$. Additional assumptions on the gap term $\mathcal{G}(\cdot)$ can be used to guarantee strict finiteness of $T^\star$.* ~~There exists at least one optimal diffusion time $T^\star$ in the interval $[0, \infty]$, which maximizes the ELBO, that is $T^\star = \arg\max_T \mathcal{L}_{\mathrm{ELBO}}(\widehat{\boldsymbol{s}}_{\boldsymbol{\theta}}, T)$.~~

It is trivial to verify that, since the optimal gap term $\mathcal{G}(\widehat{\boldsymbol{s}}_{\boldsymbol{\theta}}, T)$ is a non decreasing function in $T$ (Lemma 2), we have $\frac{\partial \mathcal{G}}{\partial T} \geq 0$. Then, we study the sign of the KL derivative, which is always negative as shown in the Appendix. Moreover, we know that that $\lim_{T \to \infty} \frac{\partial \mathrm{KL}}{\partial T} = 0$. Consequently, the function $\frac{\partial \mathcal{L}_{\mathrm{ELBO}}}{\partial T} = \frac{\partial \mathcal{G}}{\partial T} + \frac{\partial \mathrm{KL}}{\partial T}$ has at least one zero in its domain $\mathbb{R}^+$. To guarantee a stricter bounding of $T^\star$, we could study asymptotically the growth rates of $\mathcal{G}$ and the KL terms for large $T$. The investigation is technically involved and outside the scope of this paper. Nevertheless, as discussed hereafter, the numerical investigation carried out in this work suggests finiteness of $T^\star$.

~~While the proof for the general case is available in the Appendix, the analytic solution for the optimal diffusion time is elusive, as a full characterization of the gap term is particularly challenging. Additional assumptions would guarantee boundedness of $T^\star$.~~

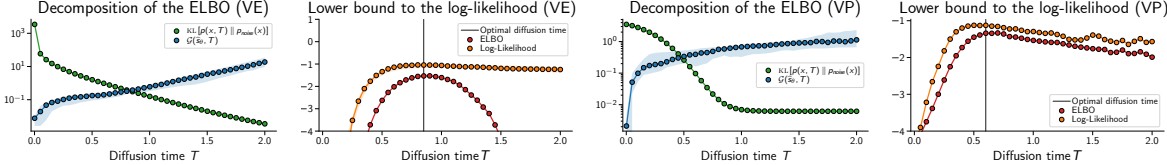

**Figure 2:** ELBO decomposition, ELBO and likelihood for a 1D toy model, as a function of diffusion time $T$. Tradeoff and optimality numerical results confirm our theory.

Empirically, we use Fig. 2 to illustrate the tradeoff and the optimality arguments through the lens of the same toy example we use in § 1. On first and third column, we show the ELBO decomposition. We can verify that $\mathcal{G}(\boldsymbol{s}_{\boldsymbol{\theta}}, T)$ is an increasing function of $T$, whereas the KL term is a decreasing function of $T$. Even in the simple case of a toy example, the tension between small and large values of $T$ is clear. On the second and fourth, we show the values of the ELBO and of the likelihood as a function of $T$. We then verify the validity of our claims: the ELBO is neither maximized by an infinite diffusion time, nor by a "sufficiently large" value. Instead, there exists an optimal diffusion time which, for this example, is smaller than what is typically used in practical implementations, i.e. $T = 1.0$

In the Appendix , we show that optimizing the ELBO to obtain an optimal diffusion time $T^\star$ is technically feasible, without resorting to exhaustive grid search. In § 3, we present a new method that admits much smaller diffusion times, and show that the ELBO of our approach is at least as good as the one of a standard diffusion model, configured to use its optimal diffusion time $T^\star$.

### 2.4 Relation with diffusion process noise schedule

We remark that a simple modification of the noise schedule to steer the the diffusion process toward a small diffusion time (Kingma et al., 2021; Bao et al., 2022) is not a viable solution. In the Appendix, we discuss

how the optimal value of the ELBO, in the case of affine SDEs, is *invariant* to the choice of the noise schedule. Indeed, its value depends uniquely on the relative level of corruption of the initial data at the considered final diffusion time $T$, that is, the *Signal to Noise Ratio*. Naively, we could think that by selecting a twice as fast noise schedule, we would be able to obtain the same ELBO of the original schedule by diffusing only for half the time. While true, this does not provide any practical benefit in terms of computational complexity. If the noise schedule is faster, the drift terms involved in the reverse process changes more rapidly. Consequently, to *simulate* the reverse SDE with a numerical integration scheme, smaller step sizes are required to keep the same accuracy of the original noise schedule simulation. The net effect is that while the diffusion time for the continuous time dynamics is smaller, the number of integration steps is larger, with a zero net gain. The optimization of the noise schedule can however have important practical effects in terms of stability of the training and variance of the estimations, that we do not tackle in this work (Kingma et al., 2021).

## 2.5 Relation with literature on bounds and goodness of score assumptions

Few other works in the literature attempt to study the convergence properties of Diffusion models. In the work of De Bortoli et al. (2021) (Thm. 1), a total variation (TV) bound between the generated and data distribution is obtained in the form $C_1 \exp(a_1 T) + C_2 \exp(-a_2 T)$, where the constant $C_1$ depends on the maximum error over $[0, T]$ between the true and approximated score, i.e. $\max_{t \in [0,T]} \|\boldsymbol{s_\theta}(\boldsymbol{x}, t) - \nabla \log p(\boldsymbol{x}, t)\|$. In the work of De Bortoli (2022) the requirement is relaxed by setting $\max_{t \in [0,T]} \frac{\sigma^2(t)}{1 + \|x\|} \|\boldsymbol{s_\theta}(\boldsymbol{x}, t) - \nabla \log p(\boldsymbol{x}, t)\|$, where the 1-Wasserstein distance between generated and true data is bounded as $C_1 + C_2 \exp(-a_2 T) + C_3$ (Thm. 1). Other works consider the more realistic average square norm instead of the infinity norm, which is consistent with standard training of diffusion models. Moreover, Lee et al. (2022a) show how the TV bound can be expressed as a function of $\max_{t \in [0,T]} \mathbb{E}\left[\|\boldsymbol{s_\theta}(\boldsymbol{x_t}, t) - \nabla \log p(\boldsymbol{x_t}, t)\|^2\right]$ (Thms. 2.2,3.1,3.2). Related to our work, Lee et al. (2022a) find that the TV bound is optimized for a diffusion time that depends, among others, on the maximum score error. Finally, the work by Chen et al. (2022) (Thm. 2), which is concurrent to ours, shows that if $\max_{t \in [0,T]} \mathbb{E}\left[\|\boldsymbol{s_\theta}(\boldsymbol{x_t}, t) - \nabla \log p(\boldsymbol{x_t}, t)\|^2\right]$ is bounded, then the TV distance between true and generated data can be bound as $C_1 \exp(-a_1 T) + \sqrt{\epsilon T}$, plus a discretization error.

All prior approaches require assumptions on the maximum score error, which *implicitly* depends on: i) the maximum diffusion time $T$ and ii) the class of parametric score networks considered. Hence, such methods allow for the study of convergence properties, but with the following limitations. It is not clear how the score error behaves as the fitting domain ($[0, T]$) is increased, for generic class of parametric functions and generic $p_{data}$. Moreover, it is difficult to link the error assumptions with the actual training loss of diffusion models. In this work, instead, we follow a more agnostic path, as we make no assumptions about the error behavior. We notice that the optimal gap term is **always** a non decreasing function of $T$. First, we question whether current best practice for setting diffusion times are adequate: we find that in realistic implementations, diffusion times are larger than necessary. Second, we introduce an new approach, with provably the same performance of standard diffusion models but lower computational complexity, as highlighted in § 3.

# 3 A new, practical method for decreasing diffusion times

The ELBO decomposition in Eq. (9) and the bounds in Lemma 1 and Lemma 2 highlight a dilemma. We thus propose a simple method that allows us to achieve **both** a small gap $\mathcal{G}(\boldsymbol{s_\theta}, T)$, and a small discrepancy $\mathrm{KL}\left[p(\boldsymbol{x}, T) \parallel p_{noise}(\boldsymbol{x})\right]$. Before that, let us use Fig. 3 to summarize all densities involved and the effects of the various approximations, which will be useful to visualize our proposal.

The data distribution $p_{data}(\boldsymbol{x})$ is transformed into the noise distribution $p(\boldsymbol{x}, T)$ through the forward diffusion process. Ideally, starting from $p(\boldsymbol{x}, T)$ we can recover the data distribution by simulating using the exact score $\boldsymbol{\nabla} \log p$. Using the approximated score $\boldsymbol{s_\theta}$ and the same initial conditions, the backward process ends up in $q^{(1)}(\boldsymbol{x}, T)$, whose discrepancy ① to $p_{data}(\boldsymbol{x})$ is $\mathcal{G}(\boldsymbol{s_\theta}, T)$. However, the distribution $p(\boldsymbol{x}, T)$ is unknown and replaced with an easy distribution $p_{noise}(\boldsymbol{x})$, accounting for an error ⓐ measured as $\mathrm{KL}\left[p(\boldsymbol{x}, T) \,\|\, p_{noise}(\boldsymbol{x})\right]$. With score and initial distribution approximated, the backward process ends up in $q^{(3)}(\boldsymbol{x}, T)$, where the discrepancy ③ from $p_{data}$ is the sum of terms $\mathcal{G}(\boldsymbol{s_\theta}, T) + \mathrm{KL}\left[p(\boldsymbol{x}, T) \,\|\, p_{noise}\right]$.

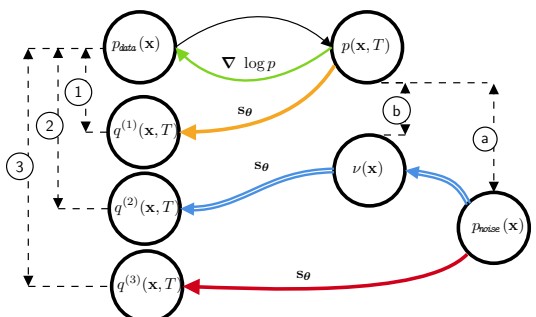

**Figure 3:** Intuitive illustration of the forward and backward diffusion processes. Discrepancies between distributions are illustrated as distances. Color coding discussed in the text.

**Multiple bridges across densities.** In a nutshell, our method allows us to reduce the gap term by selecting smaller diffusion times and by using a learned auxiliary model to transform the initial density $p_{noise}(\boldsymbol{x})$ into a density $\nu_\phi(\boldsymbol{x})$, which is as close as possible to $p(\boldsymbol{x}, T)$, thus avoiding the penalty of a large $\mathrm{KL}$ term. To implement this, we first *transform* the simple distribution $p_{noise}$ into the distribution $\nu_\phi(\boldsymbol{x})$, whose discrepancy ⓑ $\mathrm{KL}\left[p(\boldsymbol{x}, T) \,\|\, \nu_\phi(\boldsymbol{x})\right]$ is smaller than ⓐ. Then, starting from from the auxiliary model $\nu_\phi(\boldsymbol{x})$, we use the approximate score $\boldsymbol{s_\theta}$ to simulate the backward process reaching $q^{(2)}(\boldsymbol{x}, T)$. This solution has a discrepancy ② from the data distribution of $\mathcal{G}(\boldsymbol{s_\theta}, T) + \mathrm{KL}\left[p(\boldsymbol{x}, T) \,\|\, \nu_\phi(\boldsymbol{x})\right]$, which we will quantify later in the section. Intuitively, we introduce two bridges. The first bridge connects the noise distribution $p_{noise}$ to an auxiliary distribution $\nu_\phi(\boldsymbol{x})$ that is as close as possible to that obtained by the forward diffusion process. The second bridge—a standard reverse diffusion process—connects the smooth distribution $\nu_\phi(\boldsymbol{x})$ to the data distribution. Notably, our approach has important guarantees, which we discuss next.

### 3.1 Auxiliary model fitting and guarantees

We begin by stating the requirements we consider for the density $\nu_\phi(\boldsymbol{x})$. First, as it is the case for $p_{noise}$, it should be easy to generate samples from $\nu_\phi(\boldsymbol{x})$ in order to initialize the reverse diffusion process. Second, the auxiliary model should allow us to compute the likelihood of the samples generated through the overall generative process, which begins in $p_{noise}$, passes through $\nu_\phi(\boldsymbol{x})$, and arrives in $q(\boldsymbol{x}, T)$.

The fitting procedure of the auxiliary model is straightforward. First, we recognize that minimizing $\mathrm{KL}\left[p(\boldsymbol{x}, T) \,\|\, \nu_\phi(\boldsymbol{x})\right]$ w.r.t $\phi$ also minimizes $\mathbb{E}_{p(\boldsymbol{x}, T)}\left[\log \nu_\phi(\boldsymbol{x})\right]$, that we can use as loss function. To obtain the set of optimal parameters $\phi^\star$, we require samples from $p(\boldsymbol{x}, T)$, which can be easily obtained even if the density $p(\boldsymbol{x}, T)$ is not available. Indeed, by sampling from $p_{data}$, and $p(\boldsymbol{x}, T \,|\, \boldsymbol{x}_0)$, we obtain an unbiased Monte Carlo estimate of $\mathbb{E}_{p(\boldsymbol{x}, T)}\left[\log \nu_\phi(\boldsymbol{x})\right]$, and optimization of the loss can be performed. Note that due to the affine nature of the drift, the conditional distribution $p(\boldsymbol{x}, T \,|\, \boldsymbol{x}_0)$ is easy to sample from, as shown in Table 1. From a practical point of view, it is important to notice that the fitting of $\nu_\phi$ is independent from the training of the score-matching objective, i.e. the result of $I(\boldsymbol{s_\theta})$ does not depend on the shape of the auxiliary distribution $\nu_\phi$. This observation indicates that the two training procedures can be run concurrently, thus enabling considerable time savings.

Next, we show that the first bridge in our model reduces the $\mathrm{KL}$ term, even for small diffusion times.

**Proposition 3.** *Let's assume that $p_{noise}(\boldsymbol{x})$ is in the family spanned by $\nu_\phi$, i.e. there exists $\widetilde{\phi}$ such that $\nu_{\widetilde{\phi}} = p_{noise}$. Then we have that*

$$\mathrm{KL}\left[p(\boldsymbol{x}, T) \,\|\, \nu_{\phi^*}(\boldsymbol{x})\right] \leq \mathrm{KL}\left[p(\boldsymbol{x}, T) \,\|\, \nu_{\widetilde{\phi}}(\boldsymbol{x})\right] = \mathrm{KL}\left[p(\boldsymbol{x}, T) \,\|\, p_{noise}(\boldsymbol{x})\right]. \tag{10}$$

Since we introduce the auxiliary distribution $\nu$, we shall define a new ELBO for our method:

$$\mathcal{L}_{\mathrm{ELBO}}^\phi(\boldsymbol{s_\theta}, T) = \mathbb{E}_{p_{data}(\boldsymbol{x})} \log p_{data}(\boldsymbol{x}) - \mathcal{G}(\boldsymbol{s_\theta}, T) - \mathrm{KL}\left[p(\boldsymbol{x}, T) \,\|\, \nu_\phi(\boldsymbol{x})\right] \tag{11}$$

Recalling that $\widehat{\boldsymbol{s}}_{\boldsymbol{\theta}}$ is the optimal score for a generic time $T$, Proposition 3 allows us to claim that $\mathcal{L}_{\text{ELBO}}^{\boldsymbol{\phi}^{\star}}(\widehat{\boldsymbol{s}}_{\boldsymbol{\theta}}, T) \geq \mathcal{L}_{\text{ELBO}}(\widehat{\boldsymbol{s}}_{\boldsymbol{\theta}}, T)$. Then, we can state the following important result:

**Proposition 4.** *Given the existence of $T^{\star}$, defined as the diffusion time such that the ELBO is maximized (Proposition 2), there exists at least one diffusion time $\tau \leq T^{\star}$, such that $\mathcal{L}_{\text{ELBO}}^{\boldsymbol{\phi}^{\star}}(\widehat{\boldsymbol{s}}_{\boldsymbol{\theta}}, \tau) \geq \mathcal{L}_{\text{ELBO}}(\widehat{\boldsymbol{s}}_{\boldsymbol{\theta}}, T^{*})$.*

Proposition 4 has two interpretations. On the one hand, given two score models optimally trained for their respective diffusion times, our approach guarantees an ELBO that is at least as good as that of a standard diffusion model configured with its optimal time $T^{\star}$. Our method achieves this with a smaller diffusion time $\tau$, which offers sampling efficiency and generation quality. On the other hand, if we settle for an equivalent ELBO for the standard diffusion model and our approach, with our method we can afford a sub-optimal score model, that requires a smaller computational budget to be trained, while guaranteeing shorter sampling times. We elaborate on this interpretation in § 4, where our approach obtains substantial savings in terms of training iterations.

A final note is in order. The choice of the auxiliary model depends on the selected diffusion time. The larger the $T$, the "simpler" the auxiliary model can be. Indeed, the noise distribution $p(\boldsymbol{x}, T)$ approaches $p_{\textit{noise}}$, so that a simple auxiliary model is sufficient to transform $p_{\textit{noise}}$ into a distribution $\nu_{\boldsymbol{\phi}}$. Instead, for a small $T$, the distribution $p(\boldsymbol{x}, T)$ is closer to the data distribution. Then, the auxiliary model requires high flexibility and capacity. In § 4, we substantiate this discussion with numerical examples and experiments on real data.

## 3.2 Comparison with Schrödinger Bridges

In this Section, we briefly compare our method with the Schrödinger bridges approach (Chen et al., 2021b;a; De Bortoli et al., 2021), which allows one to move from an arbitrary $p_{\textit{noise}}$ to $p_{\textit{data}}$ in any finite amount of time $T$. This is achieved by simulating the SDE

$$\mathrm{d}\boldsymbol{x}_t = \left[-\boldsymbol{f}(\boldsymbol{x}_t, t') + g^2(t')\boldsymbol{\nabla} \log \hat{\psi}(\boldsymbol{x}_t, t')\right] \mathrm{d}t + g(t')\mathrm{d}\boldsymbol{w}_t, \quad \boldsymbol{x}_0 \sim p_{\textit{noise}}, \tag{12}$$

where $\hat{\psi}, \psi$ solve the Partial Differential Equation (PDE) system

$$\begin{cases} \frac{\partial \psi(\boldsymbol{x}, t)}{\partial t} = -\boldsymbol{\nabla}^{\top}\left(\psi(\boldsymbol{x}, t)\right)\boldsymbol{f}(\boldsymbol{x}, t) - \frac{g^2(t)}{2}\Delta(\psi(\boldsymbol{x}, t)), \\ \frac{\partial \hat{\psi}}{\partial t} = -\boldsymbol{\nabla}^{\top}\left(\hat{\psi}(\boldsymbol{x}, t)\boldsymbol{f}(\boldsymbol{x}, t)\right) + \frac{g^2(t)}{2}\Delta(\hat{\psi}(\boldsymbol{x}, t)), \end{cases} \quad \psi(\boldsymbol{x}, 0)\hat{\psi}(\boldsymbol{x}, 0) = p_{\textit{data}}(\boldsymbol{x}), \psi(\boldsymbol{x}, T)\hat{\psi}(\boldsymbol{x}, T) = p_{\textit{noise}}(\boldsymbol{x}). \tag{13}$$

This approach presents drawbacks compared to classical Diffusion models. First, the functions $\psi, \hat{\psi}$ are not known, and their parametric approximation is costly and complex. Second, it is much harder to obtain quantitative bounds between true and generated data as a function of the quality of such approximations.

The $\hat{\psi}, \psi$ estimation procedure simplifies considerably in the particular case where $p_{\textit{noise}}(\boldsymbol{x}) = p(\boldsymbol{x}, T)$, for arbitrary $T$. The solution of Eq. (13) is indeed $\psi(\boldsymbol{x}, t) = 1, \hat{\psi}(\boldsymbol{x}, t) = p(\boldsymbol{x}, t)$. The first PDE of the system is satisfied when $\psi$ is a constant. The second PDE is the Fokker-Planck equation, satisfied by $\hat{\psi}(\boldsymbol{x}, t) = p(\boldsymbol{x}, t)$. Boundary conditions are also satisfied. In this scenario, a sensible objective is the score-matching, as getting $\boldsymbol{\nabla} \log \hat{\psi}$ equal to the true score $\nabla \log p$ allows perfect generation.

Unfortunately, it is difficult to generate samples from $p(\boldsymbol{x}, T)$, the starting conditions of Eq. (12). A trivial solution is to select $T \to \infty$ in order to have $p_{\textit{noise}}$ as the simple and analytically known steady state distribution of Eq. (1). This corresponds to the classical diffusion models approach, which we discussed in the previous Sections. An alternative solution is to keep $T$ finite and *cover* the first part of the bridge from $p_{\textit{noise}}$ to $p(\boldsymbol{x}, T)$ with an auxiliary model. This provides a different interpretation of our method, which allows for smaller diffusion times while keeping good generative quality.

## 3.3 An extension for density estimation

Diffusion models can be also used for density estimation by transforming the diffusion SDE into an equivalent Ordinary Differential Equation (ODE) whose marginal distribution $p(\boldsymbol{x}, t)$ at each time instant coincide to

that of the corresponding SDE (Song et al., 2021c). The exact equivalent ODE requires the score $\boldsymbol{\nabla} \log p(\boldsymbol{x}_t, t)$, which in practice is replaced by the score model $\boldsymbol{s_\theta}$, leading to the following ODE

$$\mathrm{d}\boldsymbol{x}_t = \left( \boldsymbol{f}(\boldsymbol{x}_t, t) - \frac{1}{2} g(t)^2 \boldsymbol{s_\theta}(\boldsymbol{x}_t, t) \right) \mathrm{d}t \quad \text{with} \quad \boldsymbol{x}_0 \sim p_{\textit{data}} \,, \tag{14}$$

whose time varying probability density is indicated with $\widetilde{p}(\boldsymbol{x}, t)$. Note that the density $\widetilde{p}(\boldsymbol{x}, t)$, is in general **not** equal to the density $p(\boldsymbol{x}, t)$ associated to Eq. (1), with the exception of perfect score matching (Song et al., 2021b). The reverse time process is modeled as a Continuous Normaxlizing Flow (CNF) (Chen et al., 2018; Grathwohl et al., 2019) initialized with distribution $p_{\textit{noise}}(\boldsymbol{x})$; then, the likelihood of a given value $\boldsymbol{x}_0$ is

$$\log \widetilde{p}(\boldsymbol{x}_0) = \log p_{\textit{noise}}(\boldsymbol{x}_T) + \int_{t=0}^{T} \boldsymbol{\nabla} \cdot \left( \boldsymbol{f}(\boldsymbol{x}_t, t) - \frac{1}{2} g(t)^2 \boldsymbol{s_\theta}(\boldsymbol{x}_t, t) \right) \mathrm{d}t. \tag{15}$$

To use our proposed model for density estimation, we also need to take into account the ODE dynamics. We focus again on the term $\log p_{\textit{noise}}(\boldsymbol{x}_T)$ to improve the expected log likelihood. For consistency, our auxiliary density $\nu_\phi$ should now maximize $\mathbb{E}_{\sim(14)} \log \nu_\phi(\boldsymbol{x}_T)$ instead of $\mathbb{E}_{\sim(1)} \log \nu_\phi(\boldsymbol{x}_T)$. However, the simulation of Eq. (14) requires access to $\boldsymbol{s_\theta}$ which, in the endeavor of density estimation, is available only once the score model has been trained. Consequently optimization w.r.t. $\phi$ can only be performed sequentially, whereas for generative purposes it could be done concurrently. While the sequential version is expected to perform better, experimental evidence indicates that improvements are marginal, justifying the adoption of the more efficient concurrent version.

## 4 Experiments

We now present numerical results on the MNIST and CIFAR10 datasets, to support our claims in §§ 2 and 3. We follow a standard experimental setup (Song et al., 2021a;b; Huang et al., 2021; Kingma et al., 2021): we use a standard U-Net architecture with time embeddings (Ho et al., 2020) and we report the log-likelihood in terms of bit per dimension (BPD) and the Fréchet Inception Distance (FID) scores (uniquely for CIFAR10). Although the FID score is a standard metric for ranking generative models, caution should be used against over-interpreting FID improvements (Kynkäänniemi et al., 2022). Similarly, while the theoretical properties of the models we consider are obtained through the lens of ELBO maximization, the log-likelihood measured in terms of BPD should be considered with care (Theis et al., 2016). Finally, we also report the number of neural function evaluations (NFE) for computing the relevant metrics. We compare our method to the standard score-based model (Song et al., 2021c). The full description on the experimental setup is presented in Appendix.

**On the existence of $T^\star$.** We look for further empirical evidence of the existence of a $T^\star < \infty$, as stated in Proposition 2. For the moment, we shall focus on the baseline model (Song et al., 2021c), where no auxiliary models are introduced. Results are reported in Table 2. For MNIST, we observe how times $T = 0.6$ and $T = 1.0$ have comparable performance in terms of BPD, implying that any $T \geq 1.0$ is at best unnecessary and generally detrimental. Similarly, for CIFAR10, it is possible to notice that the best value of BPD is achieved for $T = 0.6$, outperforming all other values.

**Table 2:** Optimal $T$ in (Song et al., 2021c)

| Dataset | Time $T$ | bpd ($\downarrow$) |
|---------|----------|--------------------|
| MNIST   | 1.0      | 1.16               |
|         | 0.6      | **1.16**           |
|         | 0.4      | 1.25               |
|         | 0.2      | 1.75               |
| CIFAR10 | 1.0      | 3.09               |
|         | 0.6      | **3.07**           |
|         | 0.4      | 3.09               |
|         | 0.2      | 3.38               |

**Our auxiliary models.** In § 3 we introduced an auxiliary model to minimize the mismatch between initial distributions of the backward process. We now specify the family of parametric distributions we have considered. Clearly, the choice of an auxiliary model also depends on the data distribution, in addition to the choice of diffusion time $T$.

For our experiments, we consider two auxiliary models: (i) a Dirichlet process Gaussian mixture model (DPGMM) (Rasmussen, 1999; Görür & Edward Rasmussen, 2010) for MNIST and (ii) Glow Kingma & Dhariwal (2018), a flexible normalizing flow for CIFAR10. Both of them satisfy our requirements: they allow exact likelihood computation and they are equipped with a simple sampling procedure. As discussed in § 3, auxiliary model complexity should be adjusted as a function of $T$. This is confirmed experimentally in Fig. 4, where we use the number of mixture components of the DPGMM as a proxy to measure the complexity of the auxiliary model.

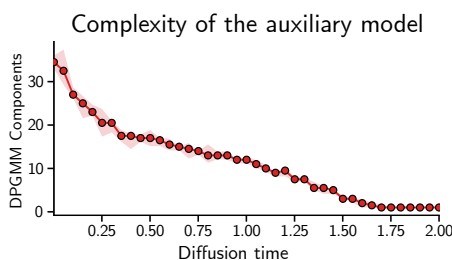

**Figure 4:** Complexity of the auxiliary model as function of diffusion time (reported median and 95 quantiles on 4 random seeds).

**Reducing $T$ with auxiliary models.** We now show how it is possible to obtain a comparable (or better) performance than the baseline model for a wide range of diffusion times $T$. For MNIST, setting $\tau = 0.4$ produces good performance both in terms of BPD (Table 3) and visual sample quality (Fig. 5). We also consider the sequential extension (S) to compute the likelihood, but remark marginal improvements compared to a concurrent implementation. Similarly for the CIFAR10 dataset, in Table 4 we observe how our method achieves better BPD than the baseline diffusion for $T = 1$. Moreover, our approach outperforms the baselines for the corresponding diffusion time in terms of FID score (additional non-curated samples in the Appendix). In Figure 10 we provide a non curated subset of qualitative results, showing that our method for a diffusion time equal to 0.4 still produces appealing images, while the vanilla approach fails. We finally notice how the proposed method has comparable performance w.r.t. several other competitors, while stressing that many orthogonal to our solutions (like diffusion in latent space (Vahdat et al., 2021), or the selection of higher order schemes (Jolicoeur-Martineau et al., 2021)) can actually be combined with our methodology.

**Training and sampling efficiency** In Fig. 7, the horizontal line corresponds to the best performance of a fully trained baseline model for $T = 1.0$ (Song et al., 2021c). To achieve the same performance of the baseline, variants of our method require fewer iterations, which translate in training efficiency. For the sake of fairness, the total training cost of our method should account for the auxiliary model training, which however can be done concurrently to the diffusion process. As an illustration for CIFAR10, using four GPUs, the baseline model requires ~ 6.4 days of training. With our method we trained the auxiliary and diffusion models for ~ 2.3 and 2 days respectively, leading to a total training time of $\max\{2.3, 2\} = 2.3$ days. Similar training curves can be obtained for the MNIST dataset, where the training time for DPGMMs is negligible.

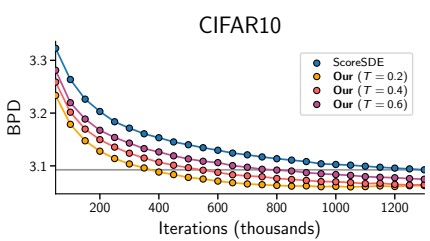

**Figure 7:** Training curves of score models for different diffusion time $T$, recorded during the span of 1.3 millions iterations.

**Figure 5:** Visualization of some samples

**Table 3:** Experiment results on MNIST. For our method, (S) is for the extension in § 3.3

| Model | nfe(↓) (ODE) | bpd (↓) | |
|---|---|---|---|
| ScoreSDE | 300 | 1.16 | |
| ScoreSDE ($T = 0.6$) | 258 | 1.16 | |
| **Our** ($T = 0.6$) | 258 | 1.16 | 1.14 (S) |
| ScoreSDE ($T = 0.4$) | 235 | 1.25 | |
| **Our** ($T = 0.4$) | 235 | 1.17 | 1.16 (S) |
| ScoreSDE ($T = 0.2$) | 191 | 1.75 | |
| **Our** ($T = 0.2$) | 191 | 1.33 | 1.31 (S) |

**Table 4:** Experimental results on CIFAR10, including other relevant baselines and sampling efficiency enhancements from the literature.

| Model | fid($\downarrow$) | bpd ($\downarrow$) | nfe ($\downarrow$) (SDE) | nfe ($\downarrow$) (ODE) |
|---|---|---|---|---|
| ScoreSDE (Song et al., 2021c) | 3.64 | 3.09 | 1000 | 221 |
| ScoreSDE ($T = 0.6$) | 5.74 | 3.07 | 600 | 200 |
| ScoreSDE ($T = 0.4$) | 24.91 | 3.09 | 400 | 187 |
| ScoreSDE ($T = 0.2$) | 339.72 | 3.38 | 200 | 176 |
| **Our** ($T = 0.6$) | 3.72 | 3.07 | 600 | 200 |
| **Our** ($T = 0.4$) | 5.44 | 3.06 | 400 | 187 |
| **Our** ($T = 0.2$) | 14.38 | 3.06 | 200 | 176 |
| ARDM (Hoogeboom et al., 2022) | – | 2.69 | 3072 | |
| VDM(Kingma et al., 2021) | 4.0 | 2.49 | 1000 | |
| D3PMs (Austin et al., 2021) | 7.34 | 3.43 | 1000 | |
| DDPM (Ho et al., 2020) | 3.21 | 3.75 | 1000 | |
| Gotta Go Fast (Jolicoeur-Martineau et al., 2021) | 2.44 | – | 180 | |
| LSGM (Vahdat et al., 2021) | 2.10 | 2.87 | 120/138 | |
| ARDM-P (Hoogeboom et al., 2022) | – | 2.68/2.74 | 200/50 | |

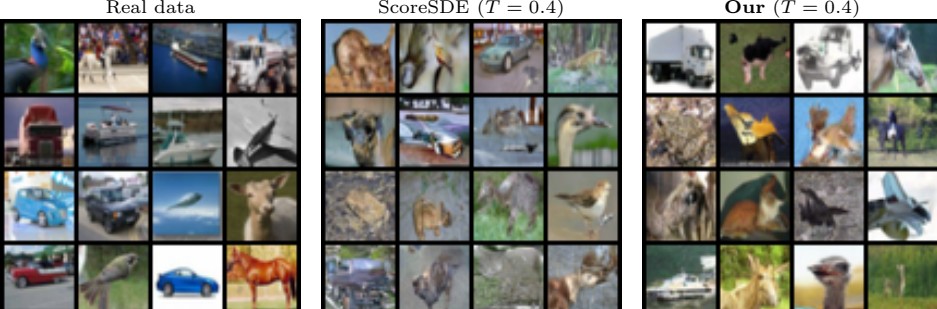

| Real data | ScoreSDE ($T = 0.4$) | **Our** ($T = 0.4$) |

**Figure 6:** Visualization of some samples on CIFAR10.

Sampling speed benefits are evident from Tables 3 and 4. When considering the SDE version of the methods the number of sampling steps can decrease linearly with $T$, in accordance with theory (Kloeden & Platen, 1995), while retaining good BPD and FID scores. Similarly, although not in a linear fashion, the number of steps of the ODE samplers can be reduced by using a smaller diffusion time $T$.

Finally, we test the proposed methodology on the more challenging CELEBA 64x64 dataset. In this case, we use a variance exploding diffusion and we consider again Glow as the auxiliary model. The results, presented in Table 6, report the log-likelihood performance of different methods (qualitative results are reported in Appendix). On the two extremes of the complexity we have the original diffusion (VE, $T = 1.0$) with the best BPD and the highest complexity, and Glow which provides a much simpler scheme with worse performance. In the table we report the BPD and the NFE metrics for smaller diffusion times, in three different configurations: naively neglecting the mismatch (ScoreSDE) or using the auxiliary model (Our). Interestingly, we found that the best results are obtained by using a combination of diffusion models pretrained for $T = 1.0$. The summary of the content of this table is the following: by accepting a small degradation in terms of BPD we can reduce the computational cost by almost one order of magnitude. We think it would be interesting to study more performing auxiliary models to improve performance of our method on challenging datasets.

## 5 Conclusion

Diffusion-based generative models emerged as an extremely competitive approach for a wide range of application domains. In practice, however, these models are resource hungry, both for their training and for sampling new data points. In this work, we have introduced the key idea of considering diffusion times $T$ as a free variable which should be chosen appropriately. We have shown that the choice of $T$ introduces a trade-off, for which an optimal "sweet spot" exists. In standard diffusion-based models, smaller values of $T$

**Table 5:** Experimental results on CELEBA 64

| Model | **bpd** (↓) (ODE) | **nfe** (↓) |
|---:|:---:|:---:|
| ScoreSDE (Song et al., 2021c) | 2.13 | 68 |
| ScoreSDE ($T = 0.5$) | 8.06 | 15 |
| ScoreSDE ($T = 0.2$) | 12.1 | 9 |
| **Our** ($T = 0.5$) | 2.48 | 16 |
| **Our** ($T = 0.2$) | 2.58 | 9 |
| **Our** with pretrain diffusion ($T = 0.5$) | 2.36 | 16 |
| **Our** with pretrain diffusion ($T = 0.2$) | 2.32 | 9 |
| Glow (Kingma & Dhariwal, 2018) | 3.74 | 1 |

are preferable for efficiency reasons, but sufficiently large $T$ are required to reduce approximation errors of the forward dynamics. Thus, we devised a novel method that allows for an arbitrary selection of diffusion times, where even small values are allowed. Our method closes the gap between practical and ideal diffusion dynamics, using an auxiliary model. Our empirical validation indicated that the performance of our approach was comparable and often superior to standard diffusion models, while being efficient both in training and sampling.

**Limitations.** In this work, the experimental protocol has been defined to corroborate our methodological contribution, and not to achieve state-of-the-art performance. A more extensive empirical evaluation of model architectures, sampling methods, and additional datasets could benefit practitioners in selecting an appropriate configuration of our method. An additional limitation is the descriptive, and not prescriptive, nature of Proposition 2: we know that $T^\star$ exists, but an explicit expression to identify the optimal diffusion is out of reach.

## Broader Impact Statement

We inherit the same ethical concerns of all generative models, as they could be used to produce fake or misleading information to the public.

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

## A    Generic definitions and assumptions

Our work builds upon the work in Song et al. (2021b), which should be considered as a basis for the developments hereafter. In this supplementary material we use the following shortened notation for a generic $\omega > 0$:

$$\mathcal{N}_\omega(\boldsymbol{x}) \stackrel{\text{def}}{=} \mathcal{N}(\boldsymbol{x}; \boldsymbol{0}, \omega \boldsymbol{I}). \tag{16}$$

It is useful to notice that $\nabla \log(\mathcal{N}_\omega(\boldsymbol{x})) = -\frac{1}{\omega}\boldsymbol{x}$.

For an arbitrary probability density $p(\boldsymbol{x})$ we define the convolution ($*$ operator) with $\mathcal{N}_\omega$ using notation

$$p_\omega(\boldsymbol{x}) = p(\boldsymbol{x}) * \mathcal{N}_\omega(\boldsymbol{x}). \tag{17}$$

Equivalently, $p_\omega(\boldsymbol{x}) = \exp\left(\frac{\omega}{2}\Delta\right)p(\boldsymbol{x})$, and consequently $\frac{dp_\omega(\boldsymbol{x})}{d\omega} = \frac{1}{2}\Delta p(\boldsymbol{x})$, where $\Delta = \nabla^\top \nabla$. Notice that by considering the Dirac delta function $\delta(\boldsymbol{x})$, we have the equality $\delta_\omega(\boldsymbol{x}) = \mathcal{N}_\omega(\boldsymbol{x})$.

In the following derivations, we make use of the Stam–Gross logarithmic Sobolev inequality result in (Villani, 2009, p. 562 Example 21.3):

$$\text{KL}\left[p(\boldsymbol{x}) \,\|\, \mathcal{N}_\omega(\boldsymbol{x})\right] = \int p(\boldsymbol{x}) \log\left(\frac{p(\boldsymbol{x})}{\mathcal{N}_\omega(\boldsymbol{x})}\right) \mathrm{d}\boldsymbol{x} \leq \frac{\omega}{2}\int \left\|\nabla\left(\log\frac{p(\boldsymbol{x})}{\mathcal{N}_\omega(\boldsymbol{x})}\right)\right\|^2 p(\boldsymbol{x})\mathrm{d}\boldsymbol{x}. \tag{18}$$

## B    Deriving Equation (4) from Huang et al. (2021)

We start with Eq. (25) of Huang et al. (2021) which, in our notation, reads

$$\log q(\boldsymbol{x}, T) \geq \mathbb{E}\left[\log p_{noise}(\boldsymbol{x}_T) \mid \boldsymbol{x}_0 = \boldsymbol{x}\right] - \int_0^T \mathbb{E}\left[\frac{1}{2}g^2(t)\|\boldsymbol{s_\theta}(\boldsymbol{x}_t)\|^2 + \nabla^\top\left(g^2(t)\boldsymbol{s_\theta}(\boldsymbol{x}_t) - \boldsymbol{f}(\boldsymbol{x}_t, t)\right) \mid \boldsymbol{x}_0 = \boldsymbol{x}\right]\mathrm{d}t.$$

The first step is to take the expected value w.r.t $\boldsymbol{x}_0 \sim p_{data}$ on both sides of the above inequality

$$\mathbb{E}_{p_{data}}\left[\log q(\boldsymbol{x}, T)\right] \geq \mathbb{E}\left[\log p_{noise}(\boldsymbol{x}_T)\right] - \int_0^T \mathbb{E}\left[\frac{1}{2}g^2(t)\|\boldsymbol{s_\theta}(\boldsymbol{x}_t)\|^2 + \nabla^\top\left(g^2(t)\boldsymbol{s_\theta}(\boldsymbol{x}_t) - \boldsymbol{f}(\boldsymbol{x}_t, t)\right)\right]\mathrm{d}t. \tag{19}$$

We focus on rewriting the term

$$\int_0^T \mathbb{E}\left[\nabla^\top\left(g^2(t)\boldsymbol{s_\theta}(\boldsymbol{x}_t) - \boldsymbol{f}(\boldsymbol{x}_t, t)\right)\right]\mathrm{d}t = \int_0^T p(\boldsymbol{x}, t|\boldsymbol{x}_0)p_{data}(\boldsymbol{x}_0)\nabla^\top\left(g^2(t)\boldsymbol{s_\theta}(\boldsymbol{x}) - \boldsymbol{f}(\boldsymbol{x}, t)\right)\mathrm{d}\boldsymbol{x}\mathrm{d}\boldsymbol{x}_0\mathrm{d}t =$$

$$-\int_0^T \nabla^\top\left(p(\boldsymbol{x}, t|\boldsymbol{x}_0)p_{data}(\boldsymbol{x}_0)\right)\left(g^2(t)\boldsymbol{s_\theta}(\boldsymbol{x}) - \boldsymbol{f}(\boldsymbol{x}, t)\right)\mathrm{d}\boldsymbol{x}\mathrm{d}\boldsymbol{x}_0\mathrm{d}t =$$

$$-\int_0^T \left(p(\boldsymbol{x}, t|\boldsymbol{x}_0)p_{data}(\boldsymbol{x}_0)\right)^{-1}\nabla^\top\left(p(\boldsymbol{x}, t|\boldsymbol{x}_0)p_{data}(\boldsymbol{x}_0)\right)\left(g^2(t)\boldsymbol{s_\theta}(\boldsymbol{x}) - \boldsymbol{f}(\boldsymbol{x}, t)\right)\left(p(\boldsymbol{x}, t|\boldsymbol{x}_0)p_{data}(\boldsymbol{x}_0)\right)\mathrm{d}\boldsymbol{x}\mathrm{d}\boldsymbol{x}_0\mathrm{d}t =$$

$$-\int_0^T \nabla^\top\left(\log(p(\boldsymbol{x}, t|\boldsymbol{x}_0)) + \log(p_{data}(\boldsymbol{x}_0))\right)\left(g^2(t)\boldsymbol{s_\theta}(\boldsymbol{x}) - \boldsymbol{f}(\boldsymbol{x}, t)\right)\left(p(\boldsymbol{x}, t|\boldsymbol{x}_0)p_{data}(\boldsymbol{x}_0)\right)\mathrm{d}\boldsymbol{x}\mathrm{d}\boldsymbol{x}_0\mathrm{d}t =$$

$$-\int_0^T \nabla^\top\left(\log(p(\boldsymbol{x}, t|\boldsymbol{x}_0))\right)\left(g^2(t)\boldsymbol{s_\theta}(\boldsymbol{x}) - \boldsymbol{f}(\boldsymbol{x}, t)\right)\left(p(\boldsymbol{x}, t|\boldsymbol{x}_0)p_{data}(\boldsymbol{x}_0)\right)\mathrm{d}\boldsymbol{x}\mathrm{d}\boldsymbol{x}_0\mathrm{d}t =$$

$$-\int_0^T \mathbb{E}\left[\nabla^\top\left(\log(p(\boldsymbol{x}_t, t|\boldsymbol{x}_0))\right)\left(g^2(t)\boldsymbol{s_\theta}(\boldsymbol{x}_t) - \boldsymbol{f}(\boldsymbol{x}_t, t)\right)\right]\mathrm{d}t.$$

Consequently, we can rewrite the r.h.s of Equation (19) as

$$\mathbb{E}\left[\log p_{\textit{noise}}\left(\boldsymbol{x}_T\right)\right] - \int_0^T \mathbb{E}\left[\frac{1}{2}g^2(t)\|\boldsymbol{s_\theta}(\boldsymbol{x}_t)\|^2 - g^2(t)\nabla^\top\left(\log(p(\boldsymbol{x}_t,t|\boldsymbol{x}_0))\right)\boldsymbol{s_\theta}(\boldsymbol{x}) + g^2(t)\nabla^\top\left(\log(p(\boldsymbol{x}_t,t|\boldsymbol{x}_0))\right)\boldsymbol{f}(\boldsymbol{x},t)\right]dt =$$

$$\mathbb{E}\left[\log p_{\textit{noise}}\left(\boldsymbol{x}_T\right)\right] - \int_0^T \mathbb{E}\left[\frac{1}{2}g^2(t)\|\boldsymbol{s_\theta}(\boldsymbol{x}_t) - \nabla\left(\log(p(\boldsymbol{x}_t,t|\boldsymbol{x}_0))\right)\|^2\right]dt -$$

$$\frac{1}{2}\int_0^T \mathbb{E}\left[g^2(t)\|\nabla\left(\log(p(\boldsymbol{x}_t,t|\boldsymbol{x}_0))\right)\| + 2\nabla^\top\left(\log(p(\boldsymbol{x}_t,t|\boldsymbol{x}_0))\right)\boldsymbol{f}(\boldsymbol{x},t)\right]dt,$$

that is exactly Equation (4).

## C   Proof of Eq. (5)

We prove the following result

$$I(\boldsymbol{s_\theta},T) \geq \underbrace{I(\boldsymbol{\nabla}\log p, T)}_{\overset{\text{def}}{=}K(T)} = \frac{1}{2}\int_{t=0}^T g^2(t)\mathbb{E}_{\sim(1)}\left[\|\boldsymbol{\nabla}\log p(\boldsymbol{x}_t,t) - \boldsymbol{\nabla}\log p(\boldsymbol{x}_t,t\,|\,\boldsymbol{x}_0)\|\right]^2 dt.$$

*Proof.* We prove that for generic positive $\lambda(\cdot)$, and $T_2 > T_1$ the following holds:

$$\int_{t=T_1}^{T_2} \lambda(t)\mathrm{E}_{\sim(1)}\left[\|\boldsymbol{s}(\boldsymbol{x}_t,t) - \nabla\log p(\boldsymbol{x}_t,t|\boldsymbol{x}_0)\|^2\right]dt \geq \int_{t=T_1}^{T_2} \lambda(t)\mathrm{E}_{\sim(1)}\left[\|\nabla\log p(\boldsymbol{x}_t,t) - \nabla\log p(\boldsymbol{x}_t,t|\boldsymbol{x}_0)\|^2\right]dt.$$
$$(20)$$

First we compute the functional derivative (w.r.t $\boldsymbol{s}$)

$$\frac{\delta}{\delta\boldsymbol{s}}\int_{t=T_1}^{T_2}\lambda(t)\mathrm{E}_{\sim(1)}\left[\|\boldsymbol{s}(\boldsymbol{x}_t,t) - \nabla\log p(\boldsymbol{x}_t,t|\boldsymbol{x}_0)\|^2\right]dt = 2\int_{t=T_1}^{T_2}\lambda(t)\mathrm{E}_{\sim(1)}\left[(\boldsymbol{s}(\boldsymbol{x}_t,t) - \nabla\log p(\boldsymbol{x}_t,t|\boldsymbol{x}_0))\right]dt =$$

$$2\int_{t=T_1}^{T_2}\lambda(t)\mathrm{E}_{\sim(1)}\left[(\boldsymbol{s}(\boldsymbol{x}_t,t) - \nabla\log p(\boldsymbol{x}_t,t))\right]dt,$$

where we used

$$\mathrm{E}_{\sim(1)}\left[\nabla\log p(\boldsymbol{x}_t,t|\boldsymbol{x}_0)\right] = \int \nabla\log p(\boldsymbol{x},t|\boldsymbol{x}_0)p(\boldsymbol{x},t|\boldsymbol{x}_0)p_{\textit{data}}(\boldsymbol{x}_0)d\boldsymbol{x}d\boldsymbol{x}_0 =$$

$$\int \nabla p(\boldsymbol{x},t|\boldsymbol{x}_0)p_{\textit{data}}(\boldsymbol{x}_0)d\boldsymbol{x}d\boldsymbol{x}_0 = \int \nabla p(\boldsymbol{x},t)d\boldsymbol{x} = \mathrm{E}_{\sim(1)}\left[\nabla\log p(\boldsymbol{x}_t,t)\right].$$

Consequently we can obtain the optimal $\boldsymbol{s}$ through

$$\frac{\delta}{\delta\boldsymbol{s}}\int_{t=T_1}^{T_2}\lambda(t)\mathrm{E}_{\sim(1)}\left[\|\boldsymbol{s}(\boldsymbol{x}_t,t) - \nabla\log p(\boldsymbol{x}_t,t|\boldsymbol{x}_0)\|^2\right]dt = 0 \rightarrow \boldsymbol{s}(\boldsymbol{x},t) = \nabla\log p(\boldsymbol{x},t). \qquad (21)$$

Substitution of this result into Eq. (20) directly proves the desired inequality.

As a byproduct, we prove the correctness of Eq. (5), since it is a particular case of Eq. (20), with $\lambda = g^2, T_1 = 0, T_2 = T$. Since $K(T)$ is a minimum, the decomposition $I(\boldsymbol{s_\theta},T) = K(T) + \mathcal{G}(\boldsymbol{s_\theta},T)$ implies $K(T) + \mathcal{G}(\boldsymbol{s_\theta},T) \geq K(T) \rightarrow \mathcal{G}(\boldsymbol{s_\theta},T) \geq 0.$ □

# D   Proof of Proposition 1

**Proposition 1.** *Given the stochastic dynamics defined in Eq. (1), it holds that*

$$\mathbb{E}_{\sim(1)} \log p(\boldsymbol{x}_T, T) - K(T) + R(T) = \mathbb{E}_{p_{data}(\boldsymbol{x})} \log p_{data}(\boldsymbol{x}). \tag{8}$$

*Proof.* We consider the pair of equations

$$\begin{aligned}
\mathrm{d}\boldsymbol{x}_t &= \left[-\boldsymbol{f}(\boldsymbol{x}_t, t') + g^2(t')\boldsymbol{\nabla} \log q(\boldsymbol{x}_t, t)\right] \mathrm{d}t + g(t')\mathrm{d}\boldsymbol{w}(t), \\
\mathrm{d}\boldsymbol{x}_t &= \boldsymbol{f}(\boldsymbol{x}_t, t)\mathrm{d}t + g(t)\mathrm{d}\boldsymbol{w}(t),
\end{aligned} \tag{22}$$

where $t' = T - t$, $q$ is the density of the backward process and $p$ is the density of the forward process. These equations can be interpreted as a particular case of the following pair of SDEs (corresponding to Huang et al. (2021) eqn (4) and (17)[1]).

$$\mathrm{d}\boldsymbol{x}_t = \underbrace{\left[-\boldsymbol{f}(\boldsymbol{x}_t, t') + g^2(t')\boldsymbol{\nabla} \log q(\boldsymbol{x}_t, t)\right]}_{\boldsymbol{\mu}(\boldsymbol{x}_t, t)} \mathrm{d}t + \underbrace{g(t')}_{\sigma(t)} \mathrm{d}\boldsymbol{w}(t),$$

$$\mathrm{d}\boldsymbol{x}_t = \left[\underbrace{\boldsymbol{f}(\boldsymbol{x}_t, t) - g^2(t)\boldsymbol{\nabla} \log q(\boldsymbol{x}_t, t')}_{-\boldsymbol{\mu}(\boldsymbol{x}_t, t')} + \underbrace{g(t)}_{\sigma(t')} \boldsymbol{a}(\boldsymbol{x}_t, t)\right] \mathrm{d}t + g(t)\mathrm{d}\boldsymbol{w}(t), \tag{23}$$

where Eq. (22) is recovered considering $\boldsymbol{a}(\boldsymbol{x}, t) = \sigma(t')\boldsymbol{\nabla} \log q(\boldsymbol{x}, t') = g(t)\boldsymbol{\nabla} \log q(\boldsymbol{x}, t')$. Eq. (23) is associated to an ELBO (Huang et al. (2021), Thm 3) that is attained with equality if and only if $\boldsymbol{a}(\boldsymbol{x}, t) = \sigma(t')\boldsymbol{\nabla} \log q(\boldsymbol{x}, t')$. Consequently, we can write the following equality associated to the backward process of Eq. (22)

$$\log q(\boldsymbol{x}, T) = \mathbb{E}\left[-\frac{1}{2}\int_0^T ||\boldsymbol{a}(\boldsymbol{x}_t, t)||^2 + 2\nabla^\top \boldsymbol{\mu}(\boldsymbol{x}_t, t')ds + \log q(\boldsymbol{x}_T, 0) \ \Big| \boldsymbol{x}_0 = \boldsymbol{x}\right], \tag{24}$$

where expected value is taken w.r.t. dynamics of the associated forward process.

By careful inspection of the couple of equations we notice that in the process $\boldsymbol{x}_t$ the drift includes the $\boldsymbol{\nabla} \log q(\boldsymbol{x}_t, t)$ term, while in our main (1) we have $\boldsymbol{\nabla} \log p(\boldsymbol{x}_t, t')$. In general the two vector fields do not agree. However, if we select as starting distribution of the generating process $p(\boldsymbol{x}, T)$, i.e. $q(\boldsymbol{x}, 0) = p(\boldsymbol{x}, T)$, then $\forall t, q(\boldsymbol{x}, t) = p(\boldsymbol{x}, t')$.

Given initial conditions, the time evolution of the density $p$ is fully described by the Fokker-Planck equation

$$\frac{d}{dt}p(\boldsymbol{x}, t) = -\nabla^\top (\boldsymbol{f}(\boldsymbol{x}, t)p(\boldsymbol{x}, t)) + \frac{g^2(t)}{2}\Delta(p(\boldsymbol{x}, t)), \quad p(\boldsymbol{x}, 0) = p_{data}(\boldsymbol{x}). \tag{25}$$

Similarly, for the density $q$,

$$\frac{d}{dt}q(\boldsymbol{x}, t) = -\nabla^\top \left(-\boldsymbol{f}(\boldsymbol{x}, t')q(\boldsymbol{x}, t) + g^2(t')\boldsymbol{\nabla} \log q(\boldsymbol{x}, t)q(\boldsymbol{x}, t)\right) + \frac{g^2(t')}{2}\Delta(q(\boldsymbol{x}, t)), \quad q(\boldsymbol{x}, 0) = p(\boldsymbol{x}, T). \tag{26}$$

By Taylor expansion we have

$$q(\boldsymbol{x}, \delta t) = q(\boldsymbol{x}, 0) + \delta t \left(\frac{d}{dt}q(\boldsymbol{x}, t)\right)_{t=0} + \mathcal{O}(\delta t^2) =$$

$$q(\boldsymbol{x}, 0) + \delta t \left(-\nabla^\top \left(-\boldsymbol{f}(\boldsymbol{x}, T)q(\boldsymbol{x}, 0) + g^2(T)\boldsymbol{\nabla} \log q(\boldsymbol{x}, 0)q(\boldsymbol{x}, 0)\right) + \frac{g^2(T)}{2}\Delta(q(\boldsymbol{x}, 0))\right) + \mathcal{O}(\delta t^2) =$$

$$q(\boldsymbol{x}, 0) + \delta t \left(\nabla^\top (\boldsymbol{f}(\boldsymbol{x}, T)q(\boldsymbol{x}, 0)) - \frac{g^2(T)}{2}\Delta(q(\boldsymbol{x}, 0))\right) + \mathcal{O}(\delta t^2),$$

---

[1]Notice that our notation for the roles of $p, q$ is swapped w.r.t. Huang et al. (2021)

and

$$p(\boldsymbol{x}, T - \delta t) = p(\boldsymbol{x}, T) - \delta t \left( \frac{d}{\mathrm{d}t} p(\boldsymbol{x}, t) \right)_{t=T} + \mathcal{O}(\delta t^2) =$$

$$p(\boldsymbol{x}, T) - \delta t \left( -\nabla^\top \left( \boldsymbol{f}(\boldsymbol{x}, T) p(\boldsymbol{x}, T) \right) + \frac{g^2(T)}{2} \Delta(p(\boldsymbol{x}, T)) \right) + \mathcal{O}(\delta t^2) =$$

$$p(\boldsymbol{x}, T) + \delta t \left( \nabla^\top \left( \boldsymbol{f}(\boldsymbol{x}, T) p(\boldsymbol{x}, T) \right) - \frac{g^2(T)}{2} \Delta(p(\boldsymbol{x}, T)) \right) + \mathcal{O}(\delta t^2)$$

Since $q(\boldsymbol{x}, 0) = p(\boldsymbol{x}, T)$, we finally have $q(\boldsymbol{x}, \delta t) - p(\boldsymbol{x}, T - \delta t) = \mathcal{O}(\delta t^2)$. This holds for arbitrarily small $\delta t$. By induction, with similar reasoning, we claim that $q(\boldsymbol{x}, t) = p(\boldsymbol{x}, t')$.

This last result allows us to rewrite Eq. (22) as the pair of SDEs

$$\begin{aligned} \mathrm{d}\boldsymbol{x}_t &= \left[ -\boldsymbol{f}(\boldsymbol{x}_t, t') + g^2(t') \boldsymbol{\nabla} \log p(\boldsymbol{x}_t, t') \right] \mathrm{d}t + g(t') \mathrm{d}\boldsymbol{w}(t), \\ \mathrm{d}\boldsymbol{x}_t &= \boldsymbol{f}(\boldsymbol{x}_t, t) \mathrm{d}t + g(t) \mathrm{d}\boldsymbol{w}(t). \end{aligned} \tag{27}$$

Moreover, since $q(\boldsymbol{x}, T) = p(\boldsymbol{x}, 0) = p_{data}(\boldsymbol{x})$, together with the result Eq. (24), we have the following equality

$$\log p_{data}(\boldsymbol{x}) = \mathbb{E}\left[ -\frac{1}{2} \int_0^T ||\boldsymbol{a}(\boldsymbol{x}_t, t)||^2 + 2\nabla^\top \boldsymbol{\mu}(\boldsymbol{x}_t, t') \mathrm{d}t + \log p(\boldsymbol{x}_T, T) \quad |\boldsymbol{x}_0 = \boldsymbol{x} \right]. \tag{28}$$

Consequently

$$\mathbb{E}_{\boldsymbol{x} \sim p_{data}} \left[ \log p_{data}(\boldsymbol{x}) \right] = \mathbb{E}\left[ \log p(\boldsymbol{x}_T, T) \right] + \mathbb{E}\left[ -\frac{1}{2} \int_0^T ||\boldsymbol{a}(\boldsymbol{x}_t, t)||^2 + 2\nabla^\top \boldsymbol{\mu}(\boldsymbol{x}_t, t') \mathrm{d}t \right] =$$

$$\mathbb{E}\left[ \log p(\boldsymbol{x}_T, T) \right] + \mathbb{E}\left[ -\frac{1}{2} \int_0^T g(t)^2 ||\boldsymbol{\nabla} \log p(\boldsymbol{x}_t, t)||^2 + 2\nabla^\top \left( -\boldsymbol{f}(\boldsymbol{x}_t, t) + g^2(t) \boldsymbol{\nabla} \log p(\boldsymbol{x}_t, t) \right) \mathrm{d}t \right] =$$

$$\mathbb{E}\left[ \log p(\boldsymbol{x}_T, T) \right] + \mathbb{E}\left[ -\frac{1}{2} \int_0^T g(t)^2 ||\boldsymbol{\nabla} \log p(\boldsymbol{x}_t, t)||^2 - 2g^2(t) \boldsymbol{\nabla}_{\boldsymbol{x}}^\top \log p(\boldsymbol{x}_t, t) \boldsymbol{\nabla} \log p(\boldsymbol{x}_t, t|\boldsymbol{x}_0) \mathrm{d}t \right]$$

$$+ \mathbb{E}\left[ -\frac{1}{2} \int_0^T 2\boldsymbol{f}^\top(\boldsymbol{x}_t, t) \boldsymbol{\nabla} \log p(\boldsymbol{x}_t, t|\boldsymbol{x}_0) \mathrm{d}t \right] =$$

$$\mathbb{E}\left[ \log p(\boldsymbol{x}_T, T) \right] + \mathbb{E}\left[ -\frac{1}{2} \int_0^T g(t)^2 ||\boldsymbol{\nabla} \log p(\boldsymbol{x}_t, t) - \boldsymbol{\nabla} \log p(\boldsymbol{x}_t, t|\boldsymbol{x}_0)||^2 \mathrm{d}t \right] +$$

$$\mathbb{E}\left[ -\frac{1}{2} \int_0^T -g(t)^2 ||\boldsymbol{\nabla} \log p(\boldsymbol{x}_t, t|\boldsymbol{x}_0)||^2 + 2\boldsymbol{f}^\top(\boldsymbol{x}_t, t) \boldsymbol{\nabla} \log p(\boldsymbol{x}_t, t|\boldsymbol{x}_0) \mathrm{d}t \right].$$

Remembering the definitions

$$K(T) = \frac{1}{2} \int_{t=0}^T g^2(t) \mathbb{E}_{\sim(1)} \left[ ||\nabla \log p(\boldsymbol{x}_t, t) - \nabla \log p(\boldsymbol{x}_t, t|\boldsymbol{x}_0)|| \right]^2 \mathrm{d}t$$

$$R(T) = \frac{1}{2} \int_{t=0}^T \mathbb{E}_{\sim(1)} \left[ g^2(t) ||\nabla \log p(\boldsymbol{x}, t \,|\, \boldsymbol{x}_0)|| \right]^2 - 2\boldsymbol{f}^\top(\boldsymbol{x}, t) \boldsymbol{\nabla} \log p(\boldsymbol{x}, t \,|\, \boldsymbol{x}_0) \mathrm{d}t,$$

we finally conclude the proof that

$$\mathbb{E}_{\sim(1)}[\log p(\boldsymbol{x}_T, T)] - K(T) + R(T) = \mathbb{E}_{\boldsymbol{x} \sim p_{data}}[\log p_{data}(\boldsymbol{x})]. \tag{29}$$

$\square$

# E    Proof of Lemma 1

In this Section we prove the validity of Lemma 1 for the case of Variance Preserving (VP) and Variance Exploding (VE) SDEs. Remember, as reported also in main Table 1, that the above mentioned classes correspond to $\alpha(t) = -\frac{1}{2}\beta(t), g(t) = \sqrt{\beta(t)}, \beta(t) = \beta_0 + (\beta_1 - \beta_0)t$ and $\alpha(t) = 0, g(t) = \sqrt{\frac{d\sigma^2(t)}{dt}}, \sigma^2(t) = \left(\frac{\sigma_{max}}{\sigma_{min}}\right)^t$ respectively.

**Lemma 1.** *For the classes of SDEs considered (Table 1), the discrepancy between $p(\boldsymbol{x}, T)$ and the $p_{noise}(\boldsymbol{x})$ can be bounded as follows.*

*For Variance Preserving SDEs, it holds that:* KL $[p(\boldsymbol{x}, T) \parallel p_{noise}(\boldsymbol{x})] \leq C_1 \exp\left(-\int_0^T \beta(t)dt\right)$.

*For Variance Exploding SDEs, it holds that:* KL $[p(\boldsymbol{x}, T) \parallel p_{noise}(\boldsymbol{x})] \leq C_2 \frac{1}{\sigma^2(T)-\sigma^2(0)}$.

## E.1    The variance Preserving (VP) convergence

We associate this class of SDEs to the Fokker Planck operator

$$\mathcal{L}^{\dagger}(t) = \frac{1}{2}\beta(t)\nabla^{\top}\left(\boldsymbol{x} \cdot + \nabla(\cdot)\right), \tag{30}$$

and consequently $\frac{dp(\boldsymbol{x},t)}{dt} = \mathcal{L}^{\dagger}(t)p(\boldsymbol{x},t)$. Simple calculations show that $\lim_{T \to \infty} p(\boldsymbol{x}, T) = \mathcal{N}_1(\boldsymbol{x})$.

We compute bound the time derivative of the KL term as

$$\begin{aligned}
\frac{d}{dt}\text{KL}\left[p(\boldsymbol{x}, T) \parallel \mathcal{N}_1(\boldsymbol{x})\right] &= \int \frac{dp(\boldsymbol{x},t)}{dt}\log\left(\frac{p(\boldsymbol{x},t)}{\mathcal{N}_1(\boldsymbol{x})}\right)d\boldsymbol{x} + \int \frac{p(\boldsymbol{x},t)}{p(\boldsymbol{x},t)}\frac{dp(\boldsymbol{x},t)}{dt}d\boldsymbol{x} = \\
&\frac{1}{2}\beta(t)\int \nabla^{\top}\left(-\nabla\log(\mathcal{N}_1(\boldsymbol{x}))p(\boldsymbol{x},t)) + \nabla p(\boldsymbol{x},t))\right)\log\left(\frac{p(\boldsymbol{x},t)}{\mathcal{N}_1(\boldsymbol{x})}\right)d\boldsymbol{x} = \\
&-\frac{1}{2}\beta(t)\int p(\boldsymbol{x},t)\left(-\nabla\log(\mathcal{N}_1(\boldsymbol{x})) + \nabla\log p(\boldsymbol{x},t))\right)^{\top}\nabla(\log\left(\frac{p(\boldsymbol{x},t)}{\mathcal{N}_1(\boldsymbol{x})}\right))d\boldsymbol{x} = \\
&-\frac{1}{2}\beta(t)\int p(\boldsymbol{x},t)\nabla(\log\left(\frac{p(\boldsymbol{x},t)}{\mathcal{N}_1(\boldsymbol{x})}\right))^{\top}\nabla(\log\left(\frac{p(\boldsymbol{x},t)}{\mathcal{N}_1(\boldsymbol{x})}\right))d\boldsymbol{x} = -\frac{1}{2}\beta(t)\int p(\boldsymbol{x},t)||\nabla(\log\left(\frac{p(\boldsymbol{x},t)}{\mathcal{N}_1(\boldsymbol{x})}\right))||^2 d\boldsymbol{x} \\
&\leq -\beta(t)\text{KL}\left[p(\boldsymbol{x}, T) \parallel \mathcal{N}_1(\boldsymbol{x})\right]. \tag{31}
\end{aligned}$$

We then apply Gronwall's inequality (Villani, 2009) to $\frac{d}{dt}\text{KL}\left[p(\boldsymbol{x}, T) \parallel \mathcal{N}_1(\boldsymbol{x})\right] \leq -\beta(t)\text{KL}\left[p(\boldsymbol{x}, T) \parallel \mathcal{N}_1(\boldsymbol{x})\right]$ to claim

$$\text{KL}\left[p(\boldsymbol{x}, T) \parallel \mathcal{N}_1(\boldsymbol{x})\right] \leq \text{KL}\left[p(\boldsymbol{x}, 0) \parallel \mathcal{N}_1(\boldsymbol{x})\right]\exp\left(-\int_0^T \beta(s)ds\right). \tag{32}$$

To claim validity of the result, we need to assume that $p(\boldsymbol{x}, t)$ has finite first and second order derivatives, and that KL $[p(\boldsymbol{x}, 0) \parallel \mathcal{N}_1(\boldsymbol{x})] < \infty$.

### E.2 The Variance Exploding (VE) convergence

The first step is to bound the derivative w.r.t to $\omega$ of the divergence $\mathrm{KL}\left[p_\omega(\boldsymbol{x}) \parallel \mathcal{N}_\omega(\boldsymbol{x})\right]$, i.e.

$$
\begin{aligned}
\frac{d}{d\omega}\mathrm{KL}\left[p_\omega(\boldsymbol{x}) \parallel \mathcal{N}_\omega(\boldsymbol{x})\right] &= \int \frac{dp_\omega(\boldsymbol{x})}{d\omega}\log\left(\frac{p_\omega(\boldsymbol{x})}{\mathcal{N}_\omega(\boldsymbol{x})}\right)\mathrm{d}\boldsymbol{x} + \int \frac{p_\omega(\boldsymbol{x})}{p_\omega(\boldsymbol{x})}\frac{dp_\omega(\boldsymbol{x})}{d\omega}\mathrm{d}\boldsymbol{x} - \int \frac{p_\omega(\boldsymbol{x})}{\mathcal{N}_\omega(\boldsymbol{x})}\frac{d\mathcal{N}_\omega(\boldsymbol{x})}{d\omega}\mathrm{d}\boldsymbol{x} = \\
&\frac{1}{2}\int (\Delta p_\omega(\boldsymbol{x}))\log\left(\frac{p_\omega(\boldsymbol{x})}{\mathcal{N}_\omega(\boldsymbol{x})}\right) - (\Delta\mathcal{N}_\omega(\boldsymbol{x}))\frac{p_\omega(\boldsymbol{x})}{\mathcal{N}_\omega(\boldsymbol{x})}\mathrm{d}\boldsymbol{x} = \\
&\frac{1}{2}\int \nabla^\top\left(p_\omega(\boldsymbol{x})\nabla\log p_\omega(\boldsymbol{x})\right)\log\left(\frac{p_\omega(\boldsymbol{x})}{\mathcal{N}_\omega(\boldsymbol{x})}\right) - \nabla^\top\left(\mathcal{N}_\omega(\boldsymbol{x})\nabla\log\mathcal{N}_\omega(\boldsymbol{x})\right)\frac{p_\omega(\boldsymbol{x})}{\mathcal{N}_\omega(\boldsymbol{x})}\mathrm{d}\boldsymbol{x} = \\
&-\frac{1}{2}\int \left(p_\omega(\boldsymbol{x})\nabla\log p_\omega(\boldsymbol{x})\right)^\top \nabla(\log\left(\frac{p_\omega(\boldsymbol{x})}{\mathcal{N}_\omega(\boldsymbol{x})}\right)) - \left(\mathcal{N}_\omega(\boldsymbol{x})\nabla\log\mathcal{N}_\omega(\boldsymbol{x})\right)^\top \nabla(\frac{p_\omega(\boldsymbol{x})}{\mathcal{N}_\omega(\boldsymbol{x})})\mathrm{d}\boldsymbol{x} = \\
&-\frac{1}{2}\int \left(p_\omega(\boldsymbol{x})\nabla\log p_\omega(\boldsymbol{x})\right)^\top \nabla(\log\left(\frac{p_\omega(\boldsymbol{x})}{\mathcal{N}_\omega(\boldsymbol{x})}\right)) - \left(p_\omega(\boldsymbol{x})\nabla\log\mathcal{N}_\omega(\boldsymbol{x})\right)^\top \nabla(\log\left(\frac{p_\omega(\boldsymbol{x})}{\mathcal{N}_\omega(\boldsymbol{x})}\right))\mathrm{d}\boldsymbol{x} = \\
&-\frac{1}{2}\int p_\omega(\boldsymbol{x})||\nabla(\log\left(\frac{p_\omega(\boldsymbol{x})}{\mathcal{N}_\omega(\boldsymbol{x})}\right))||^2\mathrm{d}\boldsymbol{x} \leq -\frac{1}{\omega}\mathrm{KL}\left[p_\omega(\boldsymbol{x}) \parallel \mathcal{N}_\omega(\boldsymbol{x})\right].
\end{aligned}
\tag{33}
$$

Consequently, using again Gronwall inequality, for all $\omega_1 > \omega_0 > 0$ we have

$$
\begin{aligned}
\mathrm{KL}\left[p_{\omega_1}(\boldsymbol{x}) \parallel \mathcal{N}_{\omega_1}(\boldsymbol{x})\right] &\leq \mathrm{KL}\left[p_{\omega_0}(\boldsymbol{x}) \parallel \mathcal{N}_{\omega_0}(\boldsymbol{x})\right]\exp(-(\log(\omega_1) - \log(\omega_0))) = \\
&\mathrm{KL}\left[p_{\omega_0}(\boldsymbol{x}) \parallel \mathcal{N}_{\omega_0}(\boldsymbol{x})\right]\omega_0\frac{1}{\omega_1}.
\end{aligned}
$$

This can be directly applied to obtain the bound for VE SDE. Consider $\omega_1 = \sigma^2(T) - \sigma^2(0)$ and $\omega_0 = \sigma^2(\tau) - \sigma^2(0)$ for an arbitrarily small $\tau < T$. Then, since for the considered class of variance exploding SDE we have $p(\boldsymbol{x}, T) = p_{\sigma^2(T) - \sigma^2(0)}(\boldsymbol{x})$

$$
\mathrm{KL}\left[p(\boldsymbol{x}, T) \parallel \mathcal{N}_{\sigma^2(T) - \sigma^2(0)}(\boldsymbol{x})\right] \leq C\frac{1}{\sigma^2(T) - \sigma^2(0)}
\tag{34}
$$

where $C = \mathrm{KL}\left[p(\boldsymbol{x}, \tau) \parallel \mathcal{N}_{\sigma^2(\tau) - \sigma^2(0)}(\boldsymbol{x})\right](\sigma^2(\tau) - \sigma^2(0))$.

Similarly to the previous case, we assume that $p(\boldsymbol{x}, t)$ has finite first and second order derivatives, and that $C < \infty$.

## F  Proof of Lemma 2

**Lemma 2.** *The optimal score gap term $\mathcal{G}(\widehat{\boldsymbol{s}}_{\boldsymbol{\theta}}, T)$ is a non-decreasing function in $T$. That is, given $T_2 > T_1$, and $\boldsymbol{\theta}_1 = \arg\min_{\boldsymbol{\theta}} I(\boldsymbol{s}_{\boldsymbol{\theta}}, T_1), \boldsymbol{\theta}_2 = \arg\min_{\boldsymbol{\theta}} I(\boldsymbol{s}_{\boldsymbol{\theta}}, T_2)$, then $\mathcal{G}(\boldsymbol{s}_{\boldsymbol{\theta}_2}, T_2) \geq \mathcal{G}(\boldsymbol{s}_{\boldsymbol{\theta}_1}, T_1)$.*

*Proof.* For $\boldsymbol{\theta}_1$ defined as in the lemma, $I(\boldsymbol{s}_{\boldsymbol{\theta}_1}, T_1) = K(T_1) + \mathcal{G}(\boldsymbol{s}_{\boldsymbol{\theta}_1}, T_1)$. Next, select $T_2 > T_1$. Then, for a generic $\boldsymbol{\theta}$, including $\boldsymbol{\theta}_2$,

$$
\begin{aligned}
I(\boldsymbol{s}_{\boldsymbol{\theta}}, T_2) = \underbrace{\int_{t=0}^{T_1} g^2(t)\mathbb{E}_{\sim(1)}\left[||\boldsymbol{s}_{\boldsymbol{\theta}}(\boldsymbol{x}_t, t) - \boldsymbol{\nabla}\log p(\boldsymbol{x}_t, t|\boldsymbol{x}_0)||^2\right]\mathrm{d}t}_{=I(\boldsymbol{s}_{\boldsymbol{\theta}}, T_1) \geq K(T_1) + \mathcal{G}(\boldsymbol{s}_{\boldsymbol{\theta}_1}, T_1) = I(\boldsymbol{s}_{\boldsymbol{\theta}_1}, T_1)} + \\
\underbrace{\int_{t=T_1}^{T_2} g^2(t)\mathbb{E}_{\sim(1)}\left[||\boldsymbol{s}_{\boldsymbol{\theta}}(\boldsymbol{x}_t, t) - \boldsymbol{\nabla}\log p(\boldsymbol{x}_t, t|\boldsymbol{x}_0)||^2\right]\mathrm{d}t}_{\geq \int_{t=T_1}^{T_2} g^2(t)\mathbb{E}_{\sim(1)}[||\boldsymbol{\nabla}\log p(\boldsymbol{x}_t, t) - \boldsymbol{\nabla}\log p(\boldsymbol{x}_t, t|\boldsymbol{x}_0)||^2]\mathrm{d}t = K(T_2) - K(T_1)} \geq \mathcal{G}(\boldsymbol{s}_{\boldsymbol{\theta}_1}, T_1) + K(T_2),
\end{aligned}
$$

from which $\mathcal{G}(\boldsymbol{s}_{\boldsymbol{\theta}}, T_2) = I(\boldsymbol{s}_{\boldsymbol{\theta}}, T_2) - K(T_2) \geq \mathcal{G}(\boldsymbol{s}_{\boldsymbol{\theta}_1}, T_1)$. $\qquad\square$

## G  Proof of Proposition 2

**Proposition 2.** *Consider the ELBO decomposition in Eq. (9). We study it as a function of time $T$, and seek its optimal argument $T^\star = \arg\max_T \mathcal{L}_{\text{ELBO}}(\widehat{s}_{\boldsymbol{\theta}}, T)$. Then, the optimal diffusion time $T^\star \in \mathbb{R}^+$, and thus not necessarily $T^\star = \infty$. Additional assumptions on the gap term $\mathcal{G}(\cdot)$ can be used to guarantee strict finiteness of $T^\star$. ~~There exists at least one optimal diffusion time $T^\star$ in the interval $[0, \infty]$, which maximizes the ELBO, that is $T^\star = \arg\max_T \mathcal{L}_{\text{ELBO}}(\widehat{s}_{\boldsymbol{\theta}}, T)$.~~*

*Proof.* It is trivial to verify that since the optimal gap term $\mathcal{G}(\widehat{s}_{\boldsymbol{\theta}}, T)$ is an increasing function in $T$ Lemma 2, then $\frac{\partial \mathcal{G}}{\partial T} \geq 0$. Then, we study the sign of the KL derivative, which is always negative as shown by Eq. (31) and Eq. (33) (where we also notice $\frac{d}{dt} = \frac{d\omega}{dt}\frac{d}{d\omega}$ keep the sign). Moreover, we know that that $\lim_{T \to \infty} \frac{\partial \text{KL}}{\partial T} = 0$. Then, the function $\frac{\partial \mathcal{L}_{\text{ELBO}}}{\partial T} = \frac{\partial \mathcal{G}}{\partial T} + \frac{\partial \text{KL}}{\partial T}$ has at least one zero in $[0, \infty]$. $\qquad \square$

## H  Optimization of $T^\star$

It is possible to treat the diffusion time $T$ as an hyper-parameter and perform gradient based optimization jointly with the score model parameters $\theta$. Indeed, simple calculations show that

$$\frac{\partial \mathcal{L}_{\text{ELBO}}(\boldsymbol{s}_{\boldsymbol{\theta}}, T)}{\partial T} = \mathbb{E}\left[ \left( \boldsymbol{f}^\top(\boldsymbol{x}_T, T)\boldsymbol{\nabla} + g^2(T)\Delta \right) \log p_{\text{noise}}(\boldsymbol{x}_T) \right] + \tag{35}$$

$$- \frac{1}{2}\mathbb{E}\left[ \|\boldsymbol{s}_{\boldsymbol{\theta}}(\boldsymbol{x}_T, T) - \boldsymbol{\nabla}\log p(\boldsymbol{x}_T, T \,|\, \boldsymbol{x}_0)\|^2 \right] + \tag{36}$$

$$\frac{1}{2}\mathbb{E}\left[ g^2(T)\|\boldsymbol{\nabla}\log p(\boldsymbol{x}_T, T \,|\, \boldsymbol{x}_0)\|^2 - 2\boldsymbol{f}^\top(\boldsymbol{x}_T, T)\boldsymbol{\nabla}\log p(\boldsymbol{x}_T, T \,|\, \boldsymbol{x}_0) \right] \tag{37}$$

## I  Proof of Proposition 4

**Proposition 4.** *Given the existence of $T^\star$, defined as the diffusion time such that the ELBO is maximized (Proposition 2), there exists at least one diffusion time $\tau \leq T^\star$, such that $\mathcal{L}_{\text{ELBO}}^{\phi^\star}(\widehat{s}_{\boldsymbol{\theta}}, \tau) \geq \mathcal{L}_{\text{ELBO}}(\widehat{s}_{\boldsymbol{\theta}}, T^*)$.*

*Proof.* Since $\forall T$ we have $\mathcal{L}_{\text{ELBO}}^{\phi}(\boldsymbol{s}_{\boldsymbol{\theta}}, T) \geq \mathcal{L}_{\text{ELBO}}(\boldsymbol{s}_{\boldsymbol{\theta}}, T)$, there exists a countable set of intervals $\mathcal{I}$ contained in $[0, T^\star]$ of variable supports, where $\mathcal{L}_{\text{ELBO}}^{\phi}$ is greater than $\mathcal{L}_{\text{ELBO}}(\boldsymbol{s}_{\boldsymbol{\theta}}, T)$. Assuming continuity of $\mathcal{L}_{\text{ELBO}}^{\phi}$, in these intervals is possible to find at least one $\tau \leq T^\star$ where $\mathcal{L}_{\text{ELBO}}^{\phi^\star}(\widehat{s}_{\boldsymbol{\theta}}, \tau) \geq \mathcal{L}_{\text{ELBO}}(\widehat{s}_{\boldsymbol{\theta}}, T^*)$. $\qquad \square$

We notice that the degenerate case $\mathcal{I} = T^\star$ is obtained only when $\forall T \leq T^\star, \text{KL}\left[ p(\boldsymbol{x}, T) \,\|\, \nu_{\boldsymbol{\phi}^*}(\boldsymbol{x}) \right] = \text{KL}\left[ p(\boldsymbol{x}, T) \,\|\, p_{\text{noise}}(\boldsymbol{x}) \right]$. We expect this condition to never occur in practice.

## J  Invariance to Noise Schedule

We here discuss about the claims made in § 2.4 about the invariance of the ELBO to the particular choice of noise schedule. First in Appendix J.1 we explain how different SDEs corresponding to different noise schedules can be translated one into the other. We introduce the concept of signal-to-noise ratio (SNR). We clarify the unified score parametrization used in practice in the literature Karras et al. (2022); Kingma et al. (2021). Then, in Appendix J.2, we prove how the single elements of the ELBO depend only on the value of the SNR at the final diffusion time $T$, as claimed in the main paper.

### J.1  Preliminaries

We consider as reference SDE a pure Wiener process diffusion,

$$d\boldsymbol{x}_t = d\boldsymbol{w}_t \quad \text{with} \quad \boldsymbol{x}_0 \sim p_{\text{data}}, \tag{38}$$

It is easily seen that the solution of the random process admits representation

$$\boldsymbol{x}_t = \boldsymbol{x}_0 + \sqrt{t}\boldsymbol{\epsilon}, \quad \boldsymbol{\epsilon} \sim \mathcal{N}(\boldsymbol{0}, \boldsymbol{I}) \tag{39}$$

In this case the time varying probability density, that we indicate with $\psi$, satisfies

$$\psi(\boldsymbol{x}, t) = \exp\left(\frac{t}{2}\Delta\right)p_{\textit{data}}(\boldsymbol{x}), \quad \psi(\boldsymbol{x}, t \,|\, \boldsymbol{x}_0) = \exp\left(\frac{t}{2}\Delta\right)\delta(\boldsymbol{x} - \boldsymbol{x}_0) \tag{40}$$

Simple calculations show that

$$\nabla \log \psi(\boldsymbol{x}, \sigma^2) = \frac{\mathbb{E}[\boldsymbol{x}_0 \,|\, \boldsymbol{x}_0 + \sigma\boldsymbol{\epsilon} = \boldsymbol{x}] - \boldsymbol{x}}{\sigma^2} \stackrel{\text{def}}{=} \frac{\boldsymbol{d}(\boldsymbol{x}; \sigma^2) - \boldsymbol{x}}{\sigma^2}, \tag{41}$$

where again $\boldsymbol{x}_0 \sim p_{\textit{data}}$ and the function $\boldsymbol{d}$ can be interpreted as a *denoiser*.

Our goal is to show the relationship between equations like Equation (1), and Equation (38). In particular, we focus on *affine* SDEs, as classically done with Diffusion models. The class of considered affine SDEs is the following:

$$\mathrm{d}\boldsymbol{x}_t = \alpha(t)\boldsymbol{x}_t \mathrm{d}t + g(t)\mathrm{d}\boldsymbol{w}_t \quad \text{with} \quad \boldsymbol{x}_0 \sim p_{\textit{data}}, \tag{42}$$

In this simple linear case the process admits representation

$$\boldsymbol{x}_t = k(t)\boldsymbol{x}_0 + \sigma(t)\boldsymbol{\epsilon}, \quad \boldsymbol{\epsilon} \sim \mathcal{N}(\boldsymbol{0}, \boldsymbol{I}) \tag{43}$$

where $k(t) = \exp\left(\int\limits_0^t \alpha(s)ds\right), \sigma^2(t) = k^2(t)\int\limits_0^t \frac{g^2(s)}{k^2(s)}ds$. We can rewrite Equation (43) as $\boldsymbol{x}_t = k(t)(\boldsymbol{x}_0 + \tilde{\sigma}(t)\boldsymbol{\epsilon})$, and define the SNR as $\tilde{\sigma}(t) = \frac{\sigma(t)}{k(t)}$. The density associated to Equation (42) can be expressed as a function of $\psi$ as follows

$$p(\boldsymbol{x}, t) = k(t)^{-D}\left[\exp\left(\frac{\tilde{\sigma}^2(t)}{2}\Delta\right)p_{\textit{data}}(\boldsymbol{x})\right]_{\frac{\boldsymbol{x}}{k(t)}} = k(t)^{-D}\psi(\frac{\boldsymbol{x}}{k(t)}, \tilde{\sigma}^2(t)). \tag{44}$$

The score function associated to Equation (43) has consequently expression

$$\nabla_{\boldsymbol{x}}\log p(\boldsymbol{x}, t) = \nabla_{\boldsymbol{x}}\log\psi(\frac{\boldsymbol{x}}{k(t)}, \tilde{\sigma}^2(t)) = \frac{1}{k(t)}\nabla_{\frac{\boldsymbol{x}}{k(t)}}\log\psi(\frac{\boldsymbol{x}}{k(t)}, \tilde{\sigma}^2(t)) = \frac{k(t)\boldsymbol{d}(\frac{\boldsymbol{x}}{k(t)}; \tilde{\sigma}^2(t)) - \boldsymbol{x}}{\sigma^2(t)}. \tag{45}$$

## J.2 Different Noise schedules

Consider a diffusion of the form Equation (38) and a score network $\bar{\boldsymbol{s}}_{\boldsymbol{\theta}}$ that approximate the true score. Inspecting Equation (45), we parametrize the score network associated to a generic diffusion Equation (42) as a function of the score of the reference diffusion. The score parametrization considered in Kingma et al. (2021), can be generalized to arbitrary SDEs Karras et al. (2022). In particular, as suggested by Equation (41), we select

$$\bar{\boldsymbol{s}}_{\boldsymbol{\theta}}(\boldsymbol{x}, t) = \frac{k(t)\boldsymbol{d}_\theta(\frac{\boldsymbol{x}}{k(t)}; \tilde{\sigma}^2(t)) - \boldsymbol{x}}{\sigma^2(t)} \tag{46}$$

We proceed by showing that the different components of the ELBO depends on the diffusion time $T$ only through $\tilde{\sigma}(T)$, but not on $k(t), \sigma(t)$ singularly for any time $t < T$.

**Theorem 1.** *Consider a generic diffusion Equation (42) and parametrize the score network as $\bar{\boldsymbol{s}}_{\boldsymbol{\theta}}(\frac{\boldsymbol{x}}{k(t)}, \tilde{\sigma}(t))$. Then, the gap term $\mathcal{G}(\bar{\boldsymbol{s}}_{\boldsymbol{\theta}}, T)$ associated to Equation (42) for a diffusion time $T$ depends only on $\tilde{\sigma}(T)$ but not on $k(t), \sigma(t)$ singularly for any time $t < T$.*

*Proof.* We first rearrange the gap term

$$2\mathcal{G}(\bar{\boldsymbol{s}}_{\boldsymbol{\theta}}, T) = \int\limits_{t=0}^T g^2(t)\mathbb{E}_{\sim(42)}\left[\|\bar{\boldsymbol{s}}_{\boldsymbol{\theta}}(\boldsymbol{x}_t, t) - \boldsymbol{\nabla}\log p(\boldsymbol{x}_t, t \,|\, \boldsymbol{x}_0)\|^2\right]\mathrm{d}t -$$

$$\int\limits_{t=0}^{T} g^2(t) \mathbb{E}_{\sim (42)} \left[ \| \boldsymbol{\nabla} \log p(\boldsymbol{x}_t, t) - \boldsymbol{\nabla} \log p(\boldsymbol{x}_t, t \,|\, \boldsymbol{x}_0) \|^2 \right] \mathrm{d}t =$$

$$\int\limits_{t=0}^{T} g^2(t) \mathbb{E}_{\sim (42)} \left[ \| \bar{\boldsymbol{s}}_{\boldsymbol{\theta}}(\boldsymbol{x}_t, t) - \boldsymbol{\nabla} \log p(\boldsymbol{x}_t, t) \|^2 \right] \mathrm{d}t$$

as shown in $[]^2$. Then

$$\int\limits_{t=0}^{T} g^2(t) \| \bar{\boldsymbol{s}}_{\boldsymbol{\theta}}(\boldsymbol{x}, t) - \boldsymbol{\nabla} \log p(\boldsymbol{x}, t) \|^2 p(\boldsymbol{x}, t \,|\, \boldsymbol{x}_0) p_{data}(\boldsymbol{x}_0) \mathrm{d}\boldsymbol{x}\mathrm{d}\boldsymbol{x}_0\mathrm{d}t =$$

$$\int\limits_{t=0}^{T} g^2(t) \left\| \frac{k(t)\boldsymbol{d}_\theta(\frac{\boldsymbol{x}}{k(t)}; \tilde{\sigma}^2(t)) - \boldsymbol{x}}{\sigma^2(t)} - \frac{k(t)\boldsymbol{d}_\theta(\frac{\boldsymbol{x}}{k(t)}; \tilde{\sigma}^2(t)) - \boldsymbol{x}}{\sigma^2(t)} \right\|^2 p(\boldsymbol{x}, t \,|\, \boldsymbol{x}_0) p_{data}(\boldsymbol{x}_0) \mathrm{d}\boldsymbol{x}\mathrm{d}\boldsymbol{x}_0\mathrm{d}t =$$

$$\int\limits_{t=0}^{T} g^2(t) \left\| \frac{k(t)\boldsymbol{d}_\theta(\frac{\boldsymbol{x}}{k(t)}; \tilde{\sigma}^2(t)) - k(t)\boldsymbol{d}(\frac{\boldsymbol{x}}{k(t)}; \tilde{\sigma}^2(t))}{\sigma^2(t)} \right\|^2 p(\boldsymbol{x}, t \,|\, \boldsymbol{x}_0) p_{data}(\boldsymbol{x}_0) \mathrm{d}\boldsymbol{x}\mathrm{d}\boldsymbol{x}_0\mathrm{d}t =$$

$$\int\limits_{t=0}^{T} \frac{g^2(t)}{k^2(t)} \left\| \frac{\boldsymbol{d}_\theta(\frac{\boldsymbol{x}}{k(t)}; \tilde{\sigma}^2(t)) - \boldsymbol{d}(\frac{\boldsymbol{x}}{k(t)}; \tilde{\sigma}^2(t))}{\tilde{\sigma}^2(t)} \right\|^2 p(\boldsymbol{x}, t \,|\, \boldsymbol{x}_0) p_{data}(\boldsymbol{x}_0) \mathrm{d}\boldsymbol{x}\mathrm{d}\boldsymbol{x}_0\mathrm{d}t =$$

$$\int\limits_{t=0}^{T} \frac{g^2(t)}{k^2(t)} \left\| \frac{\boldsymbol{d}_\theta(\frac{\boldsymbol{x}}{k(t)}; \tilde{\sigma}^2(t)) - \boldsymbol{d}(\frac{\boldsymbol{x}}{k(t)}; \tilde{\sigma}^2(t))}{\tilde{\sigma}^2(t)} \right\|^2 \psi(\frac{\boldsymbol{x}}{k(t)}, \tilde{\sigma}^2(t) \,|\, \boldsymbol{x}_0) p_{data}(\boldsymbol{x}_0) k(t)^{-D} \mathrm{d}\boldsymbol{x}\mathrm{d}\boldsymbol{x}_0\mathrm{d}t =$$

subst. $\quad \tilde{\boldsymbol{x}} = \dfrac{\boldsymbol{x}}{k(t)}, \quad \mathrm{d}\tilde{\boldsymbol{x}} = \mathrm{d}\boldsymbol{x} k^{-D}(t)$

$$\int\limits_{t=0}^{T} \frac{g^2(t)}{k^2(t)} \left\| \frac{\boldsymbol{d}_\theta(\tilde{\boldsymbol{x}}; \tilde{\sigma}^2(t)) - \boldsymbol{d}(\tilde{\boldsymbol{x}}; \tilde{\sigma}^2(t))}{\tilde{\sigma}^2(t)} \right\|^2 \psi(\tilde{\boldsymbol{x}}, \tilde{\sigma}^2(t) \,|\, \boldsymbol{x}_0) p_{data}(\boldsymbol{x}_0) \mathrm{d}\tilde{\boldsymbol{x}}\mathrm{d}\boldsymbol{x}_0\mathrm{d}t =$$

subst. $\quad r = \tilde{\sigma}^2(t), \quad \mathrm{d}r = \dfrac{g^2(t)}{k^2(t)} \mathrm{d}t$

$$\int\limits_{t=0}^{\tilde{\sigma}^2(T)} \| \bar{\boldsymbol{s}}_{\boldsymbol{\theta}}(\tilde{\boldsymbol{x}}, r) - \boldsymbol{\nabla} \log \psi(\tilde{\boldsymbol{x}}, r \,|\, \boldsymbol{x}_0) \|^2 \psi(\tilde{\boldsymbol{x}}, r) p_{data}(\boldsymbol{x}_0) \mathrm{d}\tilde{\boldsymbol{x}}\mathrm{d}\boldsymbol{x}_0\mathrm{d}r$$

For any $k(t), \sigma(t)$ such that $\tilde{\sigma}(T)$ is the same, the score matching loss is the same $\qquad \square$

**Theorem 2.** *Suppose that for any $\boldsymbol{\phi}$ of the auxiliary model $\nu_{\boldsymbol{\phi}}(\boldsymbol{x})$ it exists one $\boldsymbol{\phi}'$ such that $\nu_{\boldsymbol{\phi}'}(\boldsymbol{x}) = k^{-D}\nu_{\boldsymbol{\phi}}(\frac{\boldsymbol{x}}{k})$, for any $k > 0$. Notice that this condition is trivially satisfied if the considered parametric model has the expressiveness to multiply its output by the scalar $k$. Then the minimum of Kullback-Leibler divergence betweeen $p(\boldsymbol{x}, T)$ associated to a generic diffusion Equation (42) and the density of an auxiliary model $\nu_{\boldsymbol{\phi}}(\boldsymbol{x})$ depends only on $\tilde{\sigma}(T)$ and not on $\sigma(T)$ alone.*

---

$^2$Citation not included to avoid breaking anonymity

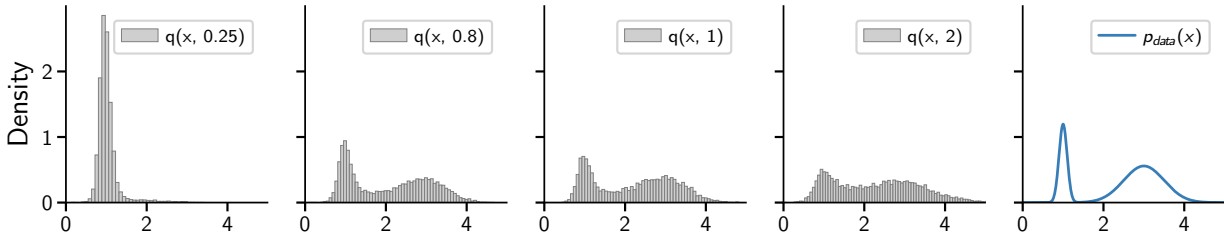

**Figure 8:** Visualization of few samples at different diffusion times $T$.

*Proof.* We start with the equality

$$\text{KL}\left[p(\boldsymbol{x},T) \parallel \nu_\phi(\boldsymbol{x})\right] = \text{KL}\left[k(T)^{-D}\psi(\frac{\boldsymbol{x}}{k(T)}, \tilde{\sigma}(T)) \parallel \nu_\phi(\boldsymbol{x})\right] = \text{KL}\left[k(T)^{-D}\psi(\frac{\boldsymbol{x}}{k(T)}, \tilde{\sigma}(T)) \parallel k(T)^{-D}\nu_{\phi'}(\frac{\boldsymbol{x}}{k(T)})\right] =$$

$$\int k(T)^{-D}\psi(\frac{\boldsymbol{x}}{k(T)}, \tilde{\sigma}(T))\log\left(\frac{\psi(\frac{\boldsymbol{x}}{k(T)}, \tilde{\sigma}(T))}{\nu_{\phi'}(\frac{\boldsymbol{x}}{k(T)})}\right)\mathrm{d}\boldsymbol{x} = \int \psi(\tilde{\boldsymbol{x}}, \tilde{\sigma}(T))\log\left(\frac{\psi(\tilde{\boldsymbol{x}}, \tilde{\sigma}(T))}{\nu_{\phi'}(\tilde{\boldsymbol{x}})}\right)\mathrm{d}\tilde{\boldsymbol{x}} =$$

$$\text{KL}\left[\psi(\boldsymbol{x}, \tilde{\sigma}(T)) \parallel \nu_{\phi'}(\boldsymbol{x})\right]$$

Then the minimimum only depends on $\tilde{\sigma}(T)$, as it is always possible to achieve the same value independently on the SDE by rescaling the auxiliary model output.

$\square$

## K  Experimental details

We here give some additional details concerning the experimental (§ 4) settings.

### K.1  Toy example details

In the toy example, we use 8192 samples from a simple Gaussian mixture with two components as target $p_{data}(\boldsymbol{x})$. In detail, we have $p_{data}(\boldsymbol{x}) = \pi\mathcal{N}(1, 0.1^2) + (1-\pi)\mathcal{N}(3, 0.5^2)$, with $\pi = 0.3$. The choice of Gaussian mixture allows to write down explicitly the time-varying density

$$p(\boldsymbol{x}_t, t) = \pi\mathcal{N}(1, s^2(t) + 0.1^2) + (1-\pi)\mathcal{N}(3, s^2(t) + 0.5^2), \tag{47}$$

where $s^2(t)$ is the marginal variance of the process at time $t$. We consider a variance exploding SDE of the type $\mathrm{d}\boldsymbol{x}_t = \sigma^t\mathrm{d}\boldsymbol{w}_t$, which corresponds to $s^2(t) = \frac{\sigma^{2t}-1}{2\log\sigma}$.

### K.2  § 4 details

We considered Variance Preserving SDE with default $\beta_0, \beta_1$ parameter settings. When experimenting on CIFAR10 we considered the NCSN++ architecture as implemented in Song et al. (2021c). Training of the score matching network has been carried out with the default set of optimizers and schedulers of Song et al. (2021c), independently of the selected $T$.

For the MNIST dataset we reduced the architecture by considering 64 features, ch_mult $= (1, 2)$ and attention resolutions equal to 8. The optimizer has been selected as the one in the CIFAR10 experiment but the warmup has been reduced to 1000 and the total number of iterations to 65000.

### K.3  Varying $T$

We clarify about the $T$ truncation procedure during both training and testing. The SDE parameters are kept unchanged irrespective of $T$. During training, as evident from Eq. (3), it is sufficient to sample randomly

the diffusion time from distribution $\mathcal{U}(0, T)$ where $T$ can take any positive value. For testing (sampling) we simply modified the algorithmic routines to begin the reverse diffusion processes from a generic $T$ instead of the default 1.0.

## L   Non curated samples

We provide for completeness collection of non curated samples for the CIFAR10 Figs. 9 to 12, MNIST dataset Figs. 13 to 16 and CELEBAdataset Fig. 17 and Table 6

**Table 6:** CELEBA: FID scores for our method and baseline ($T = 1.0$)

| Model | fid ($\downarrow$) | nfe ($\downarrow$) (SDE) |
|---|---|---|
| ScoreSDE Song et al. (2021c) | 3.90 | 1000 |
| **Our** ($T = 0.5$) | 8.06 | 500 |
| **Our** ($T = 0.2$) | 86.9 | 200 |
| **Our** with pretrain diffusion ($T = 0.5$) | 8.58 | 500 |
| **Our** with pretrain diffusion ($T = 0.2$) | 86.7 | 200 |

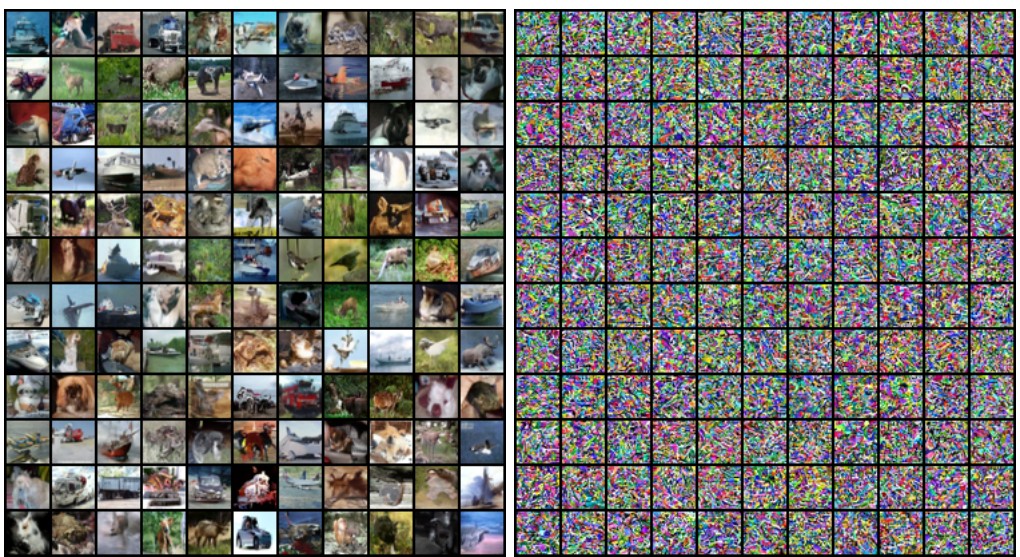

**Figure 9:** CIFAR10:Our(left) and Vanilla(right) method at $T = 0.2$

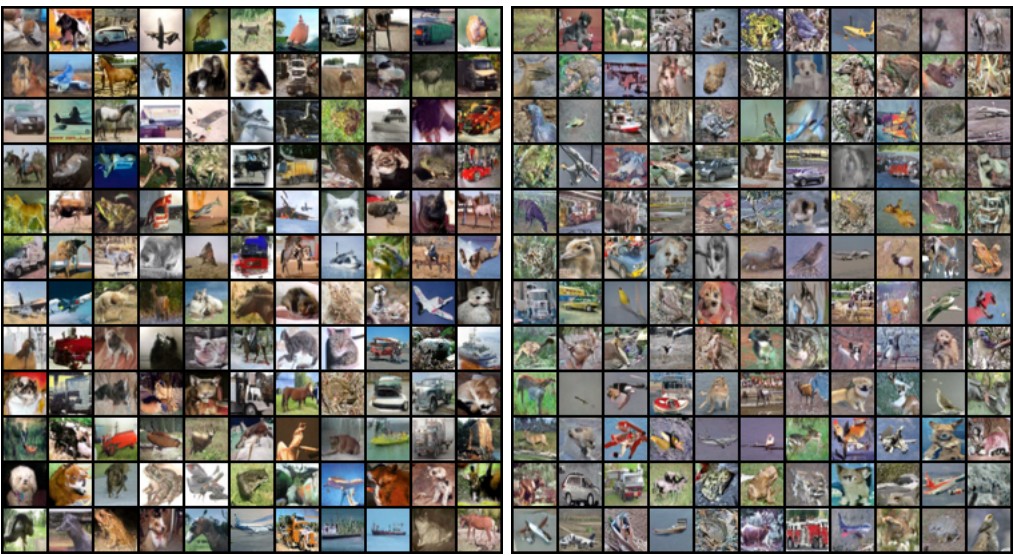

**Figure 10:** CIFAR10:Our(left) and Vanilla(right) method at $T = 0.4$

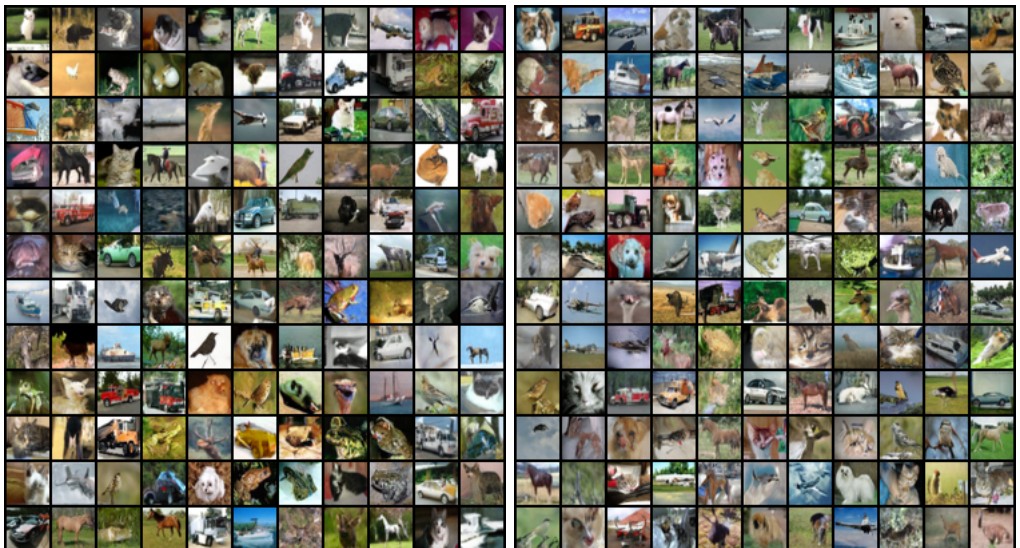

**Figure 11:** CIFAR10:Our(left) and Vanilla(right) method at $T = 0.6$

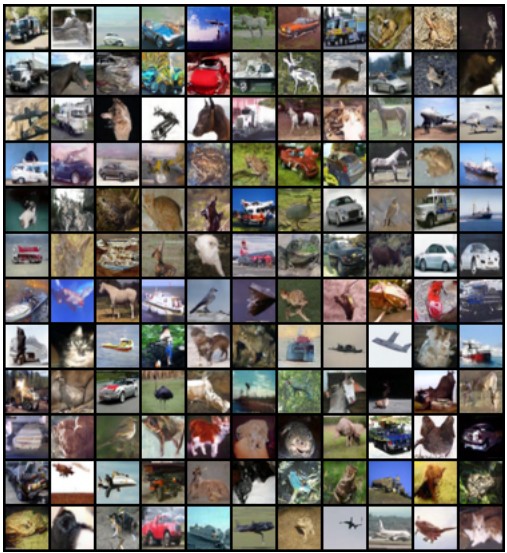

**Figure 12:** Vanilla method at $T = 1.0$

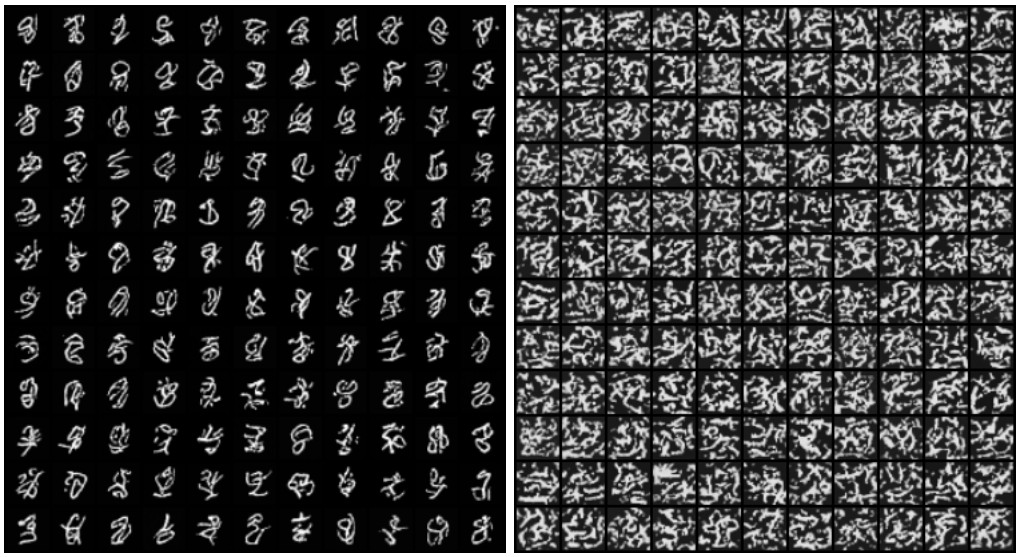

**Figure 13:** MNIST:Our(left) and Vanilla(right) method at $T = 0.2$

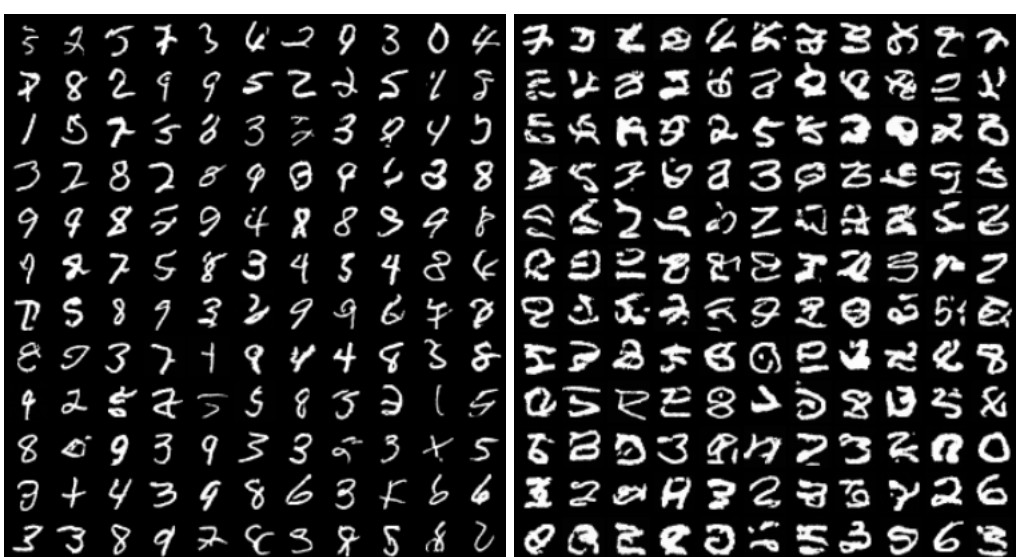

**Figure 14:** MNIST:Our(left) and Vanilla(right) method at $T = 0.4$

**Figure 15:** MNIST:Our(left) and Vanilla(right) method at $T = 0.6$

**Figure 16:** MNIST: Vanilla method at $T = 1.0$

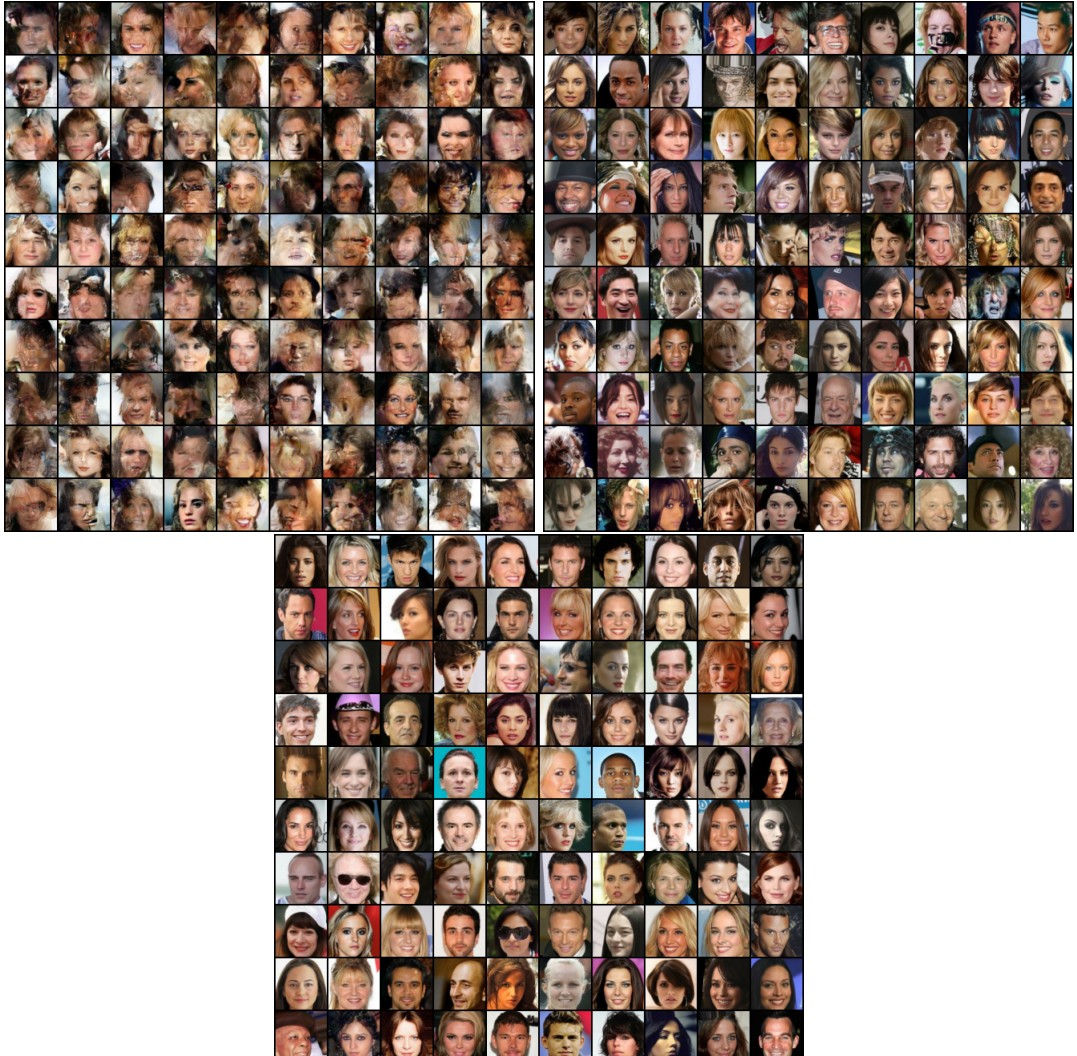

**Figure 17:** CELEBAimages. Top Left: our method with pretrained score model and Glow ($T = 0.2$), Top Right: our method with pretrained score model and Glow ($T = 0.5$) and Bottom: baseline diffusion ($T = 1.0$)

