# OpenReview forum: "Taming Diffusion Times in Score-based Generative Models: Trade-offs and Solutions"
_TMLR — Rejected by TMLR_

### Review · Reviewer_8QHN · 2022-10-24

**Summary Of Contributions:**

This paper studies the effect of diffusion time has on score-based models. The paper proposes to learn a parametric surrogate for the empirically destroyed density, which is a sensible and surprisingly has not been considered before. This is a good although incremental contribution. Second, the paper decomposes the diffusion loss and shows theoretically that the optimal total diffusion time is actually not infinite, but strikes a balance over the regularising and fit terms of ELBO. The results demonstrate the claims sufficiently. It seems that optimal time is not that small for the vanilla model, while the auxiliary bridge is much more helpful. Both are useful and interesting findings.

**Audience:**

Yes

**Claims And Evidence:**

Yes

**Requested Changes:**

The paper is generally well written (especially sections 1 and 3), however section 2 is frankly in a poor state, and needs a major rewrite.

* In q(x,t) the time variable “t” is reverse time, but this wasn’t defined. This is confusing as q(x,T) is claimed to be close to data. It would be clearer if one would use eg q(x,t’) instead, or perhaps go back to using q(x,0) to denote a fully reversed density.
* It is not explained where eq 3 is coming from. This is not the score loss, but instead one of the later variants. The p(x,t|x0) looks “magic” from the presentation alone. Given that there are many losses that have been introduced, it would be good to position this loss a bit more to literature. It would also be good to discuss if the results of this paper are dependent on this particular model variant, or if the results also generalise to other variants.
* It’s unclear where x0 is coming from in eq 3. Is there an outer E_{x0} expectation missing, since one should iterate the loss over all observations. I assume this is merged into the E_{1} thingy [please write this explicitly, it might also be useful to move to \int notation if \E gets too cumbersome], but that should then expand into two expectations: one over the starting point, another over time. I would then expect that one should replace the E_1 with E_x0 alone (since we already integrate over time).
* Fig1 y-axis should perhaps be log likelihood
* It would be good write the p_noise explicitly, I believe it has a fixed form given eq 1.
* Can you explain why we start from cross-entropy instead of evidence in eq 4? Usually ELBOs are evidence lower bounds, but this seems a CE bound.
* It’s unclear how does the overall loss eq 3 connect to the elbo of eq 4. These seem very different. Please make the connection explicit.
* The sec 2.1. is the “meat” of the paper, and yet I couldn’t follow because of insufficient explanations. The problem starts immediately at eq 4: None of this is explained. We have a cross-entropy, an ELBO, and the exotic ~1 expectation and R and I terms. The cross-entropy is not explained, so I’m not sure where it’s coming from. The ELBO is undefined, and does not match eq 3, so I’m unsure where it comes from. The log noise term seems to be another cross-entropy. The I is not explained, but seems to be score loss, and R is also not explained, but seems to be some kind of reverse stepping error. Later the K and gap are undefined, and we also get more entropies emerging in eqs 8,9 (why?). Please rewrite this section such that the equations can be understood without first reading Huang’21.
* After all the eq 9 seems sensible. I wonder where it comes from: is this novel or from Huang’21?
* In eq 9 the entropy seems constant: do we need this term?
* Is lemma 2 novel?
* There is likely a typo in table 1 at sigmaT-sigma0I.
* I can’t follow the gap discussion in 2.2 and 2.3. because gap was never defined: not sure what it is.
* I’m a bit surprised that the equations contain (cross)entropies several times, yet they are not acknowledged in text almost at all. It would be good to discuss the role of (cross)entropies in the text.
* I don’t understand fig 4. So blue is entropy that goes up and down over time. Red is number of mixture components which goes down as diffusion time increases (?). Do the red and blue curves have something to do with each other?


**Strengths And Weaknesses:**

Paper is generally clearly presented (except section 2). The problem definition is clear, methods are principled and results are appropriate.

The main weakness is the poor presentation of section 2 that is almost illegible. This needs to be revised.

---

> ### Author Response · Authors · 2022-10-31
> **Rebuttal Reviewer 8QHN**
>
> We thank the reviewer for a timely review of our paper. We are pleased to learn that the reviewer thinks our work is a valid and interesting contribution to the literature of diffusion-based models. In what follows, we answer each comment of the reviewer, with explanations. We also directly modified the submission, and uploaded a revised version of the article with modifications, clarifications and additional explanations visible in “blue”.
>
> - *In q(x,t) the time variable “t” is reverse time, but this wasn’t defined. This is confusing as q(x,T) is claimed to be close to data. It would be clearer if one would use eg q(x,t’) instead, or perhaps go back to using q(x,0) to denote a fully reversed density.*
>
> We apologize for the typo, we were referring to q(x,t’) indeed. We fixed the text clarifying the role of reverse time.
>
> - *It is not explained where eq 3 is coming from. This is not the score loss, but instead one of the later variants. The p(x,t|x0) looks “magic” from the presentation alone. Given that there are many losses that have been introduced, it would be good to position this loss a bit more to literature. It would also be good to discuss if the results of this paper are dependent on this particular model variant, or if the results also generalise to other variants.*
>
> The loss we consider is the loss defined in [Song2021c], equation 7. To clarify exposition, we re-expressed it in more familiar terms. In the introduction, for the sake of generality we consider a generic weighting $\lambda(t)$, as in eq. (4) of [Song2021b]. Then, we focus our attention on the case $\lambda(t)=g^2(t)$. To further clarify our exposition, we also expand the notation $E_{\sim (1)}$ (see also next point).
> Finally, note also that our work is valid and applies to all variants  that appear in the literature which are based on maximum likelihood training of diffusion models.
>
> - *It’s unclear where x0 is coming from in eq 3. Is there an outer E_{x0} expectation missing, since one should iterate the loss over all observations. I assume this is merged into the E_{1} thingy [please write this explicitly, it might also be useful to move to \int notation if \E gets too cumbersome], but that should then expand into two expectations: one over the starting point, another over time. I would then expect that one should replace the E_1 with E_x0 alone (since we already integrate over time).*
>
> There are three quantities we integrate over: the time t, the initial value $x_0$, and $x_t$. The notation $E_{\sim (1)}$ means, as explained in the text, that we take expectation w.r.t. the SDE of eq. (1). The random process at time t is “obtained” as follows: first, initial conditions are sampled from $p_{data}$, then the stochastic process evolves for a time t as described by the sde. We clarified the notation in the paper to improve readability.
>
> - *Fig1 y-axis should perhaps be log likelihood*
>
> Noted. Thanks for the comment, we have fixed it in our resubmission (also in Figure 2).
>
> - *It would be good write the p_noise explicitly, I believe it has a fixed form given eq 1.*
>
>  As stated in the paper:  “for the class of SDEs we consider in this work, [pnoise] is a Gaussian distribution”. The precise form depends on $\alpha(t)$ and $g(t)$. We prefer to opt for a general  exposition, but we report in Table 1 the explicit expressions of $p_{noise}$ for common models. We hope that this is satisfactory for the reviewer.
>
> - *Can you explain why we start from cross-entropy instead of evidence in eq 4? Usually ELBOs are evidence lower bounds, but this seems a CE bound.*
>
> As noted by the reviewer, usually evidence lower bounds are written as a function of a single datapoint, i.e. $log ( q(x, T)) > ELBO(x)$, where $ELBO(x)$ is the value of the bound for datapoint $x$. In this work, we take the expected value w.r.t the desired density $p_{data}$ on both sides of the inequality as  $E_{p_{data}(x)} log( q(x,T)) > E_{p_{data}(x)} ELBO(x)$.
> Indeed our goal is to reason in terms of distributions, not single datapoints. A technical byproduct of this choice is the presence of cross-entropy terms in the equations. We added comments on this choice in the revised version of the manuscript. Moreover, we expand in the appendix the details of the sentence “By manipulating the elbo derived in Huang2021, Eq. (25)”, by including all the necessary steps.
>
> - *It’s unclear how does the overall loss eq 3 connect to the elbo of eq 4. These seem very different. Please make the connection explicit.*
>
> We rewrote the loss term in eq. (3) into an equivalent form. Then, we explain in the text how the $I(s_{\theta},T)$ term in eq. (4) is the loss term from eq. (3), when $\lambda(t)=g^2(t)$. This is the only $\theta$ dependent parameter and, consequently, the only one that impacts the training of the score network. We clarified all these aspects in the text of the article.

---

> > ### Comment · Reviewer_8QHN · 2022-10-31
> > **evidence clarification**
> >
> > In eq 4 the q(x,T) is the evidence of one observed datapoint. When one takes the evidence over all datapoints, I don't see why the logarithm goes inside the expectation in the eq 4 LHS. I believe that the log should be outside the expectation in the evidence, and only go inside in the ELBO.
> >
> > My reasoning here is that log evidence should be $\log p(x) = \log \int p(x|z)p(z) dz$ for an ideal joint $p(x,z)$ notation of data $x$ and latents $z$, and here the data likelihood factorises under iid as $p(x|z) = \prod p(x_n | z)$; which leads to us having the \log\int\prod order, where we can't push the \log inside, and we can't push the product outside either.
> >
> > I definitely could be wrong and missed something, would be happy to hear your thoughts.
> >
> > On the other hand, perhaps there is some KL interpretation behind your cross-entropy, eg. could it come from KL[ p_data || q(T) ]?

---

> > > ### Author Response · Authors · 2022-11-02
> > > **clarification**
> > >
> > > We thank the reviewer for the comment. As the reviewer suspected, the interpretation of the cross-entropy term should be looked for in the Kullback-Leibler between data and generated distribution. Indeed, simple rearrangement of the terms of Eq. (9) allows to move from
> > >
> > > $
> > > 	E_{p_{data}(\boldsymbol{x})}\log q(\boldsymbol{x},T) \geq E_{p_{data}(\boldsymbol{x})}\log p_{data}(\boldsymbol{x}) - \mathcal{G}(\boldsymbol{s}_{\boldsymbol{\theta}},T) - KL [p(\boldsymbol{x},T) || p_N (\boldsymbol{x})]
> > > $
> > >
> > > to
> > >
> > > $
> > > KL [p_ {data}(\boldsymbol{x}) ||  q(\boldsymbol{x},T)] \leq  \mathcal{G}(\boldsymbol{s}_{\boldsymbol{\theta}},T) + KL [p(\boldsymbol{x},T) || p_N (\boldsymbol{x})]
> > > $
> > >
> > > ($p_N$ is the p_{noise} in the paper, Markdown is not working correctly)
> > >
> > > The reviewer doubt is originated from a latent variable model interpretation (LVM) of the diffusion models , $q(\boldsymbol{x})=\int q(\boldsymbol{x}|\boldsymbol{z})q(\boldsymbol{z})d\boldsymbol{z} $. It is possible to interpret Diffusion models as LVM (like in [Huang 2021, Kingma 2021]), but it is not directly the approach that we follow here. We sketch informally a connection to help the reviewer understand the relationship. The likelihood for $M$ independent datapoints factorizes as $\prod_{m=1}^M q(\boldsymbol{x}^{m})$. The logarithm of this is equal to the sum  $\sum_{m=1}^M \log q(\boldsymbol{x}^{m})$. This quantity is related to $E_{p_{data}(\boldsymbol{x})}\log q(\boldsymbol{x})$, as for $\boldsymbol{x}^{m}\sim p_{data}$ and $M$ large enough they do "converge" to the same value (this is a loose statement). Each one of the summands $\log q(\boldsymbol{x}^{m})$ can be bounded using the LVM model interpretation and Jensen inequalities (as done in [Ho 2020]) to obtain a bound conceptually equivalent to the one we use in this work.
> > > To avoid confusions, we prefer to avoid making such connection in the paper.

---

### Review · Reviewer_oWEy · 2022-11-04

**Summary Of Contributions:**

In the paper "Taming Diffusion Times in Score-based Generative Models: Trade-offs and Solutions", the authors quantify the role of the diffusion time T in score based generative models.  Whereas existing practice works with the case of large T, the authors in this paper show that increasing T causes a tradeoff between efficiency and model quality / approximation errors.  Precisely, the authors study an ELBO decomposition, which they show has two terms that quantify this tradeoff.  The authors establish existence of optimal diffusion times and use that to motivate the development of novel diffusion models with small diffusion times.  The authors show that their novel diffusion models has competitive performance with sota methods.


**Audience:**

Yes

**Broader Impact Concerns:**

None.

**Claims And Evidence:**

Yes

**Requested Changes:**

See weakness above regarding Proposition 2.  Some modification or commentary in the paper is needed.

On page 4 there is a type logaritmic -> logarithmic


**Strengths And Weaknesses:**

Strengths
- The result about optimal diffusion times is a surprising, non-intuitive theoretical development on a system that is of significant current importance and impact.
- The result could inspire additional work by related to diffusion models in a variety of contexts
- The writing in the paper is exceptional, and the authors do a fantastic job of conveying the mathematical system they are studying, the fundamental tradeoff they are addressing, and the developments of the paper
- The experiments provide reasonable validation for the proposed approach


Weaknesses

- I am slightly confused by a detail in Proposition 2.  The theorem states that there is an optimal diffusion time T* in [0, infty] .  Is including the possibility of T*=infty a typo?  Do you mean to say [0, infty)?  If T*=infty is included in the possible T*'s, then the proposition doesn't seem to be that meaningful, as any function over t>=0 would have an optimal value that is in [0,infty]

---

> ### Author Response · Authors · 2022-11-10
> **Rebuttal Reviewer oWEy**
>
> We thank the reviewer for the effort and the positive review. We are happy to see that the reviewer appreciated the novelty of the content of the paper, supported by the experimental validation.
>
> We agree with the reviewer that the statement of Proposition 2 could be improved. As a consequence, we refined the content of the proposition, and improved the discussion, along the following lines. We are studying the evidence lower bound as a function of $T$. Contrary to common knowledge there is no guarantee that the optimal time $T^\star$ is equal to infinity, but in general, could be any positive value ($T^\star \in\mathbb{R}^+$). To claim strict finiteness of $T^\star$, auxiliary assumptions and investigations are needed, like analysis of the growth rate of the gap and KL term for large T. The study of the KL term is a relatively well known topic (we have exponential convergence to zero and many different results from the optimal transport theory). On the other hand, analysis of the gap term would require more effort as we would need to focus on certain classes of parametric functions and study the approximation error as the domain is enlarged. While interesting, this is outside the scope of our work where we : i) verify experimentally that the optimal diffusion time $T^\star$ is smaller than infinity (and indeed in certain cases it is actually smaller than the current best practice), ii) exploit the mathematical machinery developed to study the tradeoff to suggest an improved diffusion model with even smaller diffusion times

---

### Review · Reviewer_CkXW · 2022-11-29

**Summary Of Contributions:**

In this paper, the authors provide a theoretical analysis of the role of the forward time T in denoising diffusion models. The authors advocate that there is a trade-off between the time needed for the forward diffusion to converge for which a large time T is needed and the score approximation which requires a small time T. The authors quantitfy this trade-off and propose a new method to modify the forward process. Experiments on MNIST and CIFAR10 are provided. The new proposed method suggests not to start from a Gaussian distribution but from a learned prior that approximate the end distribution of the forward process.

**Audience:**

Yes

**Broader Impact Concerns:**

None.

**Claims And Evidence:**

Yes

**Requested Changes:**

My main criticism of the paper is:

* The claim that no theory exists for the analysis of the time.

* The novelty of the ELBO which could be derived in a simpler manner.

* The lack of related work for the proposed methodology

These main concerns are detailed in the "Strength and Weaknesses" paragraph and highlighted with a (C).
Addressing these issues is critical to secure my recommendation.

**Strengths And Weaknesses:**

STRENGTHS:

* Diffusion models are a very active area of research and obtaining a better understanding of their property is a very appealing challenge. In particular the time T is of practical and theoretical importance.

* The paper is mostly well-written and the proofs seem correct to me.

* I like the idea of the auxilary model to replace the pure noise model (even though the novelty of this contribution could be discussed).

WEAKNESSES:

* (C) I strongly disagree with statements such as "While recent works have started to lay down a theoretical foundations for these models, an analytical understanding of the role of the diffusion time T is still lacking". This is not true. For example in [1, Theorem 1] there is a clear trade-off between the two terms in the upper-bound. One term corresponds to the distance between $p_T$ and the noise distribution and the other to the discretization and approximation of the backward process which includes the score approximation (we note that in the current work the discretization is not taken into account). Similarly in [2], see Theorem 4.2 for instance, the authors split the error term into a convergence term and an approximation term $\varepsilon_1$. In [3, Theorem 2] there is a clear balance between the convergence term and an approximation term (I am aware that [3] was submitted approximately at the same time as the current paper but this is not the case for [1] and [2]).

* (C) I disagree that the ELBO decomposition in 2.1 is new. In fact these results are already obtained in [4] as shown in [5, Appendix H]. I think the current way of writing the decomposition is unecessarily complicated. This can be seen as there is an explicit expression for the gap $\mathcal{G}$, see [5, Appendix H] or the proof of Theorem 1 (with the missing citation). Once we rewrite $\mathcal{G}$ like this it turns out that Equation (9) which is the key result of Section 2.1 is exactly Equation (8) in [4, Theorem 1]. I think the claims of novelty in this case are inflated.

* I am quite surprised by the FID score provided  in Table 6 as they don't seem good at all compared to the one of [6] (even though the number of NFE is twice as low).

* It is well-known that log-likelihood score do not necessarily correlate with image quality (log-likelihood is not necessarily correlated with FID score), see [7] for example in the context of diffusion models. I think the authors should discuss this discrepancy as (at least in the case of image generative modeling) the most important metric is the image quality.

* There is no confidence interval in the experimental results. Given that some of the improvement is small it is hard to know if it is negligeable or not. Some of the numbers are also quite misleading (see for example Table 2 where the authors chose to bold the second 1.16 but not the first one).

* Why do the authors use VE-SDE when they turn to the CelebA dataset. It seems to me that VP-SDE performs better.

* (C) The idea of changing the prior of the model is quite interesting I found. However I think the authors could do a better job at citing the related literature. For example [8] changes the forward kernel and does not target a Gaussian distribution. Instead a kernel with fatter tails is chosen (this is in an ODE setting). More closely related to this work, Schrodinger bridges are models based on diffusions which interpolate between arbitrary distributions [3,9,10], solving effectively the need of setting the prior distribution to be Gaussian. I think there exist deep links between the method proposed by the authors and this method. Also related is the concept of latent diffusion [11]. In the case where the encoder/decoder are invertible (normalizing flows) these can be seen as changing the forward process and the target distribution.

MINOR POINTS:

* I could not find the assumption on $p_{data}$ in Lemma 1. I may have missed something here.

* Why citation p.23 cannot be included?

* Can the influence of $g(t)$ be discussed by the authors in their setting? I have read the discussion 2.4 but I don't really see how the theory introduced by the authors fit in this discussion.

[1] Diffusion Schrödinger Bridge with Applications to Score-Based Generative Modeling -- De Bortoli, Thornton, Heng, Doucet.

[2] Convergence for score-based generative modeling with polynomial complexity -- Lee, Lu, Tan

[3] Sampling is as easy as learning the score: theory for diffusion models with minimal data assumptions -- Chen, Chewi, Li, Li, Salim, Zhang

[4] Maximum Likelihood Training of Score-Based Diffusion Models -- Song, Durkan, Ermon

[5] Convergence of denoising diffusion models under the manifold hypothesis -- De Bortoli

[6] Score-Based Generative Modeling through Stochastic Differential Equation -- Song, Sohl-Dickstein, Kingma, Kumar, Ermon, Poole

[7] Learning Fast Samplers for Diffusion Models by Differentiating Through Sample Quality -- Watson, Chan, Ho, Norouzi

[8] Poisson Flow Generative Models - Xu, Liu, Tegmark, Jaakkola

[9] Deep Generalized Schrödinger Bridge - Liu, Chen, Theodorou

[10] Solving Schrödinger Bridges via Maximum Likelihood - Vargas, Thodoroff, Lawrence, Lamacraft

[11] Score-based Generative Modeling in Latent Space - Vahdat, Kreis, Kautz

---

> ### Author Response · Authors · 2022-12-06
> **Rebuttal Reviewer CkXW**
>
> We thank the reviewer for the thorough and insightful rebuttal.
>
> We modified and clarified several sections of the paper according to the major comments (C) suggested by the reviewer. These changes appear in blue in the revised paper, where we also ``strike out'' the parts of text that we will omit from the final version of the paper.
> Hereafter, we shortly summarize those changes, and address the minor points raised by the reviewer.
>
>
>
> - *(C) I strongly disagree with statements such as "While recent works have started to lay down a theoretical foundations for these models, an analytical understanding of the role of the diffusion time T is still lacking". This is not true. For example in [1, Theorem 1] there is a clear trade-off between the two terms in the upper-bound. One term corresponds to the distance between  and the noise distribution and the other to the discretization and approximation of the backward process which includes the score approximation (we note that in the current work the discretization is not taken into account). Similarly in [2], see Theorem 4.2 for instance, the authors split the error term into a convergence term and an approximation term . In [3, Theorem 2] there is a clear balance between the convergence term and an approximation term (I am aware that [3] was submitted approximately at the same time as the current paper but this is not the case for [1] and [2]).*
>
>
> We thank the reviewer for the suggestions about the related works. We do agree that there exists relevant literature, which we should discuss more in detail in our paper.
>
> We summarize and discuss prior results in a new subsection (2.5) of the manuscript, where we also discuss the main differences between these works and ours.
> Previous works rely on assumptions about the behavior of maximum score error, for generic diffusion times $T$. This has the benefit of allowing the derivation of strict quantitative bounds between true and generated data distributions. However, it is not clear how the constants used in these assumptions could be estimated in a practical implementation of diffusion models.
> In our work we follow a different path: we investigate the existence of a trade-off in a generic setting, i.e., without any kind of assumption on the score error.
> We discuss how the overlooked trade-off result simultaneously impacts quality and computational performance of the diffusion models.
> Most importantly, to the best of our knowledge, we are the first ones to explore explicitly this trade-off on real datasets with realistic score networks.
>
>
>
> - *(C) I disagree that the ELBO decomposition in 2.1 is new. In fact these results are already obtained in [4] as shown in [5, Appendix H]. I think the current way of writing the decomposition is unecessarily complicated. This can be seen as there is an explicit expression for the gap , see [5, Appendix H] or the proof of Theorem 1 (with the missing citation). Once we rewrite  like this it turns out that Equation (9) which is the key result of Section 2.1 is exactly Equation (8) in [4, Theorem 1]. I think the claims of novelty in this case are inflated.*
>
>
> We agree with the reviewer that the writing of this part of the paper was unclear, and we apologize for the misunderstanding. Indeed, the ELBO formulation we use in our work appears in previous literature, and we now acknowledge that in the revised manuscript.
> However, we would like to stress that the contribution of our work is not the ELBO decomposition, which we derived from the result of (Huang21), but rather the novel analysis of the trade-off in choosing diffusion times in generic settings, supported by an extensive experimental validation.
>
> - *I am quite surprised by the FID score provided in Table 6 as they don't seem good at all compared to the one of [6] (even though the number of NFE is twice as low).*
>
>
> We posit that the FID score results in Table 6 have worse performance than the vanilla model as the auxiliary model we used in our experiments (Glow) has insufficient representational power.
> This is particularly true for small $T$, where the map $p_{noise}$ to $p(x,T)$ is of insufficient quality.
> The fact that NFE is smaller for smaller $T$ (Tab5, Tab6) is not surprising. NFE is indeed expected to grow with $T$, as from standard ODE/SDE numerical integration literature.
> As we stress in the main paper, the message is the following: by accepting a quality degradation we can drastically reduce the computational cost. The degradation can, in principle, become negligible by using auxiliary models with higher performance (but possibly higher costs).

---

> > ### Author Response · Authors · 2022-12-06
> > **.**
> >
> > - *It is well-known that log-likelihood score do not necessarily correlate with image quality (log-likelihood is not necessarily correlated with FID score), see [7] for example in the context of diffusion models. I think the authors should discuss this discrepancy as (at least in the case of image generative modeling) the most important metric is the image quality.*
> >
> > We are aware (and we mention this in our paper) that the log-likelihood does not necessarily correlate with the FID score, and that this latter metric has intrinsic limitations (Kynkäänniemi et al., 2022).
> > In our work we focus on a variational analysis in terms of log-likelihood of the generated distributions.
> > Experimentally, we think that an analysis in terms of Bit per dimension is more generic and less specific to the image-generation domain. For example, it is interesting to consider applications, for which diffusion models could be used, where good performance in density estimation is a primary requirement. Hence, in our work, we report FID scores to compare to previous works, but focus more on the Bit per dimension performance metric.
> >
> >
> >
> >
> >
> >
> >
> >
> >
> > - *There is no confidence interval in the experimental results. Given that some of the improvement is small it is hard to know if it is negligeable or not. Some of the numbers are also quite misleading (see for example Table 2 where the authors chose to bold the second 1.16 but not the first one).*
> >
> >
> >
> > First, we would like to note that the FID and BPD results we report in our work are obtained following the same experimental protocol of Song 2021c, including hyper-parameter settings. This allows us to considerably save on the computational budget available for our experiments. In some cases, small improvements in performance metrics are achieved with tangible reduction in diffusion times, with non-negligible consequences on training times and sampling NFEs.
> > In particular, for Table 2, we chose to bold the second 1.16 because this is achieved with a lower $T$, which is ultimately our objective.
> >
> >
> > - *Why do the authors use VE-SDE when they turn to the CelebA dataset. It seems to me that VP-SDE performs better.*
> >
> >
> > We privileged the VE-SDE to have more diverse experimental results, as the other datasets have been explored with VP-SDE, and to show that our analysis applies to other scenarios. We are aware that VP-SDE can outperform VE-SDE, and that many other architectural changes could have been implemented.
> > Our goal is however to make a comparative analysis (vanilla diffusion model, with generic SDE, versus auxiliary model method).
> >
> >
> > - *(C) The idea of changing the prior of the model is quite interesting I found. However I think the authors could do a better job at citing the related literature. For example [8] changes the forward kernel and does not target a Gaussian distribution. Instead a kernel with fatter tails is chosen (this is in an ODE setting). More closely related to this work, Schrodinger bridges are models based on diffusions which interpolate between arbitrary distributions [3,9,10], solving effectively the need of setting the prior distribution to be Gaussian. I think there exist deep links between the method proposed by the authors and this method. Also related is the concept of latent diffusion [11]. In the case where the encoder/decoder are invertible (normalizing flows) these can be seen as changing the forward process and the target distribution.*
> >
> >
> > Thank you for your observations: we added a new subsection (3.2) that clarifies this relationship. Schrodinger bridges allow one to map arbitrary distributions in finite time $T$. Their usage for generative modelling is typically limited to settings where $p_{noise}$ is a simple distribution. Their main advantage over diffusion models is that in theory they do allow to have exact mapping in finite time $T$. In practice, the estimation of the drift terms is more complex than with classical diffusion models, limiting their practical use. We now also clarify (in the revised version of our paper) how our method can be interpreted under a unified view, together with Schrodinger bridges and Diffusion models.
> >
> > Finally, we comment on latent diffusion models. In this case, the data distribution itself is transformed into a lower dimensional, latent distribution, over which classical diffusion is performed. This idea could be interpreted as complementary to our method: first, perform a reverse diffusion from noise to a latent distribution, then use an auxiliary model from the latent to the data distribution. We consider this observation as a plausible follow-up work, which we will leave for the future.
> >
> >
> >
> >
> >
> >
> >
> >
> >
> >
> > - *I could not find the assumption on  in Lemma 1. I may have missed something here.*
> >
> > We included the necessary assumptions in the Appendix. Note that those are very loose assumptions.

---

> > > ### Author Response · Authors · 2022-12-06
> > > **.**
> > >
> > > - *Why citation p.23 cannot be included?*
> > >
> > > We decided to omit from our submission the citation to avoid issues related to double blind rules. In the final version of our paper the citation will appear in full.
> > >
> > >
> > > - *Can the influence of g(t) be discussed by the authors in their setting? I have read the discussion 2.4 but I don't really see how the theory introduced by the authors fit in this discussion.*
> > >
> > >
> > > In this work we explore the trade-off on diffusion time $T$ and propose a method to reduce it, to have computational benefits.
> > > An alternative naive approach to reduce the time $T$ could be the usage of faster noise schedules. As a consequence, we discuss how the ELBO in our derivations is invariant to any change in the noise schedule. Furthermore, note that when changing the noise schedule, the simulated SDEs require finer step sizes during numerical integration, which make them more costly. In summary, the purpose of our comments in the Section is to argue that, as far as log-likelihood quality and sampling speed are concerned, changing the noise schedule is not a viable option.

---

### Decision · Action_Editors · 2023-01-29

**Recommendation:** Reject

**Comment:**

In the current version (revision), the authors claim that they found an interesting trade-off in setting the (absolute) diffusion time T for generative diffusion models.  They also claim that a bound of the likelihood is theoretically analyzed.  Based on the theoretical analysis, the authors propose to set T to a smaller value than typical values in the literature, and use a trainable distribution for the initial distribution of the reverse process.  Some empirical results are shown.

Reviewers raised concerns on clarity, novelty, and technical soundness, most of which have been addressed by the authors.  Two reviewers recommended accept, while the other leaning to reject.
I raised major concerns (see below), and asked questions to the authors and the reviewers.  The authors rebuttal didn't fully address the concerns, the two accepting reviewers didn't support the paper by addressing my concerns in one week, and the reviewer leaning to reject sympathized me.

I have to say the current paper is highly misleading and publishing it wouldn't give positive impact to the community.  Also I wan't convinced by the authors' last comments that they revise the paper so that the paper wouldn't mislead readers.  Therefore, I reject the paper.  Details follow.

My major concerns in the last communications were these three.

	1	Remove ambiguity of the setting. I didn't suggest to delve into the discretization, but suggest to make it clear that the analysis makes sense. You say you use a fixed step-size eta = T/N. But what do you mean by fixed? Fixed for different data? Fixed for different T? Why don't you say the value of N if it's fixed for all cases? Even in this communication, your answer is ambiguous. What I meant is that because of the invariance on the scale (fast diffusion for short time vs. slow diffusion for long time), if you increase T and decrease N and noise schedule is optimized for each case, you could show opposite results. My suggestion is simply to make things explicit so that readers see that the invariance can't absorb the effect of decreasing T both in SDE and ODE.
	2	No theoretical contribution can be found. You say "the first to formally and extensively discuss the tradeoff that emerges from the ELBO formulation, and corroborate observations with a thorough experimental validation." But discussion without conclusion is not theoretical contribution. Without knowledge, one expects that optimal T can be any non-negative value. The known decomposition into non-increasing and non-descreasing functions implies the trade-off, which however only applies to a bound, and also doesn't narrow the range of possible value for T. Also, this decomposition is not novel, so not the contribution of the current work. You make some statements as lemmas, which also don't narrow the range. What is the conclusion of your discussion? If you'd say you "corroborate" observations, specify which statements theoretically corroborate the observation. What unknown statements you make in the discussion, and how they corroborate observations?
	3	Not well-designed experiments I see your main claims are
	•	decreasing T from the typical setting in the literature improves the performance.
	•	Replacing the top diffusion layers with NF is beneficial. So these should be empirically supported at least. For the first point, it is unclear what is the typical setting in the literature. Probably the easiest is to take a few SOTA diffusion models, and show that shrinking T improves the performance. If the story of the paper is true, readers will expect this. Why don't you show it? For the second point, the baseline should be a carefully optimized (in terms of hyperparameters including noise schedule, step size, etc.) diffusion model. After you have an optimal diffusion model for given data, you should replace the top part with NF and show gains in terms of accuracy and/or computation time. Current experiments do not well disentangle different factors.

The first point was addressed.  The authors should clearly describe how all the parameters are set so that readers are convinced that the plots show what should be shown.
On the second point, the authors insisted nothing about the theoretical contribution in the last comments.  I suggest to change the abstract and introduction so that readers won't expect any theoretical contribution.  The current abstract says "lack of theory", which usually implies that this paper would have some contribution.  Also, it does not make sense to spend several pages with "discussion" without any conclusion or theoretical contribution.  Most of the theory parts should be removed.  For the claim the authors can make from their results, it is sufficient to show an equation that the ELBO can be decomposed into non-increasing part and non-decreasing part, and give some intuitive argument (perhaps in a few paragraphs).   In any case, the authors can't make any reasonably strong theoretical claim because they analyzed only a lower-bound and even failed to prove the existence of trade-off (optimal T is finite).  Note that any function can be decomposed into non-increasing and non-decreasing functions, so this decomposition itself says nothing about the behavior of the function at large T.
On the third point, the rebuttal was not convincing.  I suggested to show shrinking T improves "a few" sota models, because it seems that the authors assume that experts would believe that T should be set as large as possible as long as computation resource allows.  I don't really believe this assumption (the authors didn't explicitly claim but it's not appropriate to imply it).  Even the authors' chosen model, Song21, this didn't really happen.  The authors only show that, in some cases, bpd can be improved by reducing T from the original setting.  First of all, the gain in bpd is very tiny in Tables 2 and 4, and the authors even refused to do statistical test in the communication with a reviewer.  This is not acceptable, and at least the authors should show statistical significance.  A more serious problem is that shrinking T degrades FID significantly in Table 4 and 5.  Do you still say reducing T improves the performance, and recommend readers to use smaller T?  In what applications bpd is more important than FID?  This must be justified.  Don't mix up this point with computational gain, which only confuses readers on what the authors want to claim.  Lastly, in the auxiliary models, the experiments simply show that DPGMM and flow are fast and bad method, and therefore replacing the last part of diffusion model with them degrades the performance while speed up, which is not surprising.

In my understanding, what the authors can claim with the current results are

1. It was empirically shown that too large T can harm the density estimation performance.  But it shouldn't be implied that experts believe T-> infinity is the best if computation cost is ignored.  The result is not surprising, but showing a clear peak in Figure 1 can be counted as a contribution.
2. Replacing the top part of the diffusion model with simple models can speed up with marginal performance degradation.  This is again not surprising, but if the authors conduct more detailed experiments in more examples, this could be counted as a contribution.

In addition, if the authors really prove that optimal T is finite under mild conditions, this will be a theoretical contribution.  I'd suggest the authors to make 2. as the main contribution, conduct extensive experiments, and resubmit a major revision to TMLR.



**Audience:**

Yes, but the claim is wrong in the current version.  I believe that by fixing the claims and conducting further experiments, the authors can make the paper meet the TMLR requirements.

**Claims And Evidence:**

In the current version (revision), the authors claim theoretical contributions, which I couldn't find, and the authors couldn't really specify. For empirical observations, the claim should be clearer  (see the main comments below).

---

> ### Author Response · Authors · 2023-02-03
> **Feedback on the review process**
>
> While we respect the final decision by the AE we would like to share some feedback on the review process.
>
> * **Transparency**: in your final communication it is stated that reviewers expressed their decision and opinions on our work. There is no trace of this in the system, and we had no feedback whatsoever on our rebuttal (see also next point). Furthermore our interaction with the AE, where we provide our counter arguments to AE statements, are omitted from public view. A reader landing on openreview would have a partial and heavily redacted view on how we responded to critics. For this reason we take the liberty to paste below our answers to the final points raised by the AE.
>
> * **Engagement**: the key idea of TMLR and openreview is to improve the interaction with reviewers and AE. It is very important, also for transparency reasons, to have a full account of how reviewers perceive authors' rebuttals. In our case, we had no feedback, but a few words about private communications between the AE and reviewers. This is not aligned with the goals of TMLR and openreview.
>
> * **Wording**: in several points, the AE insinuates scientific misconduct. Some of these points have been concealed by setting restrictions on the visibility of the comments. Such insinuations, even when mildly alluded, are serious and should be done with evidence at hand. A research work can be easily rejected with constructive feedback, and respectful wording. Pushing a narrative that might induce the reader to believe we did something wrong, and that our work contains flaws that cannot be traced back in the review is misleading and unnecessary.
>
> * **Timeliness**: the review process should provide feedback and decisions within roughly 2 months, according to TMLR charts. In our case it took almost 4 months. The AE asked for timely answers with short notice, also to reviewers. While we did comply with all requests, we understand that reviewers may have had troubles engaging with such short notice. As the whole process was already heavily delayed, giving reviewers a bit more time to respond would have not hurt.
>
> In the future we will do our best to contribute to the TMLR community with quality work. We hope that the AE will consider our feedback and do the best to keep up with the philosophy and goals of TMLR.
>
> [our rebuttal to the AEs concern in the last communication will follow in a separate message, due to character limits]

---

> > ### Author Response · Authors · 2023-02-03
> > **Feedback on the review process, continued (1)**
> >
> > Thank you for the effort in helping us clarify our article. We understand the intention is to remove any source of ambiguity and we think our last improvements to the paper will make it a good fit to TMLR. Our answers to the comments, with clear descriptions of the modifications to the paper, are below.
> >
> > *1. Remove ambiguity of the setting. I didn't suggest to delve into the discretization, but suggest to make it clear that the analysis makes sense. You say you use a fixed step-size eta = T/N. But what do you mean by fixed? Fixed for different data? Fixed for different T? Why don't you say the value of N if it's fixed for all cases? Even in this communication, your answer is ambiguous. What I meant is that because of the invariance on the scale (fast diffusion for short time vs. slow diffusion for long time), if you increase T and decrease N and noise schedule is optimized for each case, you could show opposite results. My suggestion is simply to make things explicit so that readers see that the invariance can't absorb the effect of decreasing T both in SDE and ODE.*
> >
> > When using the SDE formulation, we proceed as follows:
> > - We set $T$ to some fixed value. For the vanilla diffusion model, we set $T=1.0$. For our approach, we set $T=0.2$, $T=0.4$, … up to $T=1.0$
> > - We set the integration step size to $\eta = 10^{-3}$
> > - Then, $N$, which is what we report as NFE in our results section, is:
> >
> >    For the vanilla diffusion model: $N=T/\eta = 1000$.
> >
> >    For our approach, $N=T/\eta=200$, $N=T/\eta=400$, … up to $N=T/\eta=1000$
> >
> > When using the ODE formulation we use the same technique described in Song 21c, which uses an adaptive step-size integrator. In this case, $\eta$ is variable, whereas $T$ is fixed. As a consequence, we count the number of integration steps $N$ and report it as NFE.
> >
> > We will add these explanations in Section 4, before presenting the results.
> >
> >
> > *2. No theoretical contribution can be found. You say "the first to formally and extensively discuss the tradeoff that emerges from the ELBO formulation, and corroborate observations with a thorough experimental validation." But discussion without conclusion is not theoretical contribution. Without knowledge, one expects that optimal T can be any non-negative value. The known decomposition into non-increasing and non-descreasing functions implies the trade-off, which however only applies to a bound, and also doesn't narrow the range of possible value for T.
> > Also, this decomposition is not novel, so not the contribution of the current work. You make some statements as lemmas, which also don't narrow the range. What is the conclusion of your discussion? If you'd say you "corroborate" observations, specify which statements theoretically corroborate the observation. What unknown statements you make in the discussion, and how they corroborate observations?*
> >
> > We acknowledge (as we already did in all our answers to reviewers) that the ELBO decomposition we use has been also shown in one previous work by Song et al. 21b. However, no reviewer nor the AE could refer to a research paper in the literature that discusses the presence of a tradeoff, and the existence of a value of $T<1.0$ that results in improved training and sampling performance. These observations are supported by:
> > - Fig 2. for both variance preserving and variance exploding SDEs
> > - Table 2, for both MNIST and CIFAR
> >
> > The conclusion of our analysis on the tradeoff is: there is no need to set a large $T$ to train diffusion models. An optimal $T$ can be smaller than $T=1.0$.
> > If – for computational budget reasons – one wishes to set $T$ smaller than the optimal value, results will deteriorate. Our proposed approach can largely recover the discrepancy measured by the KL term of the ELBO, resulting in comparable generation quality and a smaller computational cost.
> >
> > We will spell out this conclusion both in the introduction and the relevant sections. Although we already smoothed the tone in our revisions, if, according to the AE, the use of a formal language and presentation might wrongly induce the reader to believe there is a theoretical breakthrough, we are willing to rename Lemmas and Propositions to Observations.

---

> > > ### Author Response · Authors · 2023-02-03
> > > **Feedback on the review process, continued (2)**
> > >
> > > *3. Not well-designed experiments, claims should be empirically supported at least.*
> > >
> > > *Decreasing T from the typical setting in the literature improves the performance. It is unclear what is the typical setting in the literature. Probably the easiest is to take a few SOTA diffusion models, and show that shrinking T improves the performance. If the story of the paper is true, readers will expect this. Why don't you show it?*
> > >
> > > The typical setting in all previous work on diffusion models is $T=1.0$. In our experiments  we show exactly what happens when using lower values of $T$ using classical diffusion models:
> > > - In table 2, we take the diffusion model of Song et al ‘21c and show that setting $T=0.6$ achieves the same BPD performance metric, for both MNIST and CIFAR10.
> > > - In table 3, we further show that, for the diffusion model of Song et al ‘21c used on MNIST, when decreasing $T$ to $T=0.6$, the number of function evaluations for sampling decreases while maintaining the same BPD metric. If $T<0.6$, the BPD metric deteriorates, indicating that $T=0.6$ is a better choice than $T=1.0$ and $T<0.6$.
> > > - In table 4,  we zoom in on CIFAR10, and show that, for the diffusion model of Song et al ‘21c, when decreasing $T$ to $T=0.6$ we improve both the BPD metric, that goes down to 3.07, and the NFE, that goes down to 200.
> > > Similar considerations can be made for the FID score, and the corresponding number of network evaluations (NFE), where we use the backward SDE, instead of using the backward ODE. In this case, the FID score performance metric slightly deteriorates, but the number of network evaluations decreases considerably.
> > >
> > > We believe these results are what readers will look for, and we are going to spell them out in our final revision to help readers interpret the results.
> > >
> > >
> > > *Replacing the top diffusion layers with NF is beneficial. The baseline should be a carefully optimized (in terms of hyperparameters including noise schedule, step size, etc.) diffusion model. After you have an optimal diffusion model for given data, you should replace the top part with NF and show gains in terms of accuracy and/or computation time. Current experiments do not well disentangle different factors.*
> > >
> > > We were careful in configuring the baseline diffusion model from Song et al. ‘21c according to the best hyper-parameters reported by the authors, and we are confident the baseline model is working in its best conditions.
> > >
> > > Note that our approach does not alter any “layers” of a score network, but it’s rather a cascade of a diffusion model and an auxiliary one. What our approach does is to enable the training of a score-based diffusion model to happen in a more efficient regime (where the SNR is not essentially zero) and to enable the backward process to start from more favorable initial conditions, which we obtain using an auxiliary model:
> > > - In table 3, our approach achieves equal (or better, with our extension) BPD metric, while requiring fewer NFE, that is, better performance, for less effort
> > > - In table 4, our approach achieves a better BPD metric, while requiring fewer NFE, again, better performance for less effort. For the SDE case, the FID score is marginally worse (3.72 vs. 3.64) than the baseline (at $T=1.0$), but requires 600 instead of 1000 network evaluations.
> > > - In table 5, our approach achieves a slightly worse FID score, for a considerably fewer NFE, that is, comparable performance for much less effort
> > >
> > > We will spell out these observations in our final revision to help the reader navigate throughout the numerous results in our tables and figures.